# Irrigated areas drive irrigation water withdrawals

Arnald Puy [1,2 ✉], Emanuele Borgonovo[3], Samuele Lo Piano[4], Simon A. Levin [1] & Andrea Saltelli [5]

A sustainable management of global freshwater resources requires reliable estimates of the water demanded by irrigated agriculture. This has been attempted by the Food and Agriculture Organization (FAO) through country surveys and censuses, or through Global Models, which compute irrigation water withdrawals with sub-models on crop types and calendars, evapotranspiration, irrigation efficiencies, weather data and irrigated areas, among others. Here we demonstrate that these strategies err on the side of excess complexity, as the values reported by FAO and outputted by Global Models are largely conditioned by irrigated areas and their uncertainty. Modelling irrigation water withdrawals as a function of irrigated areas yields almost the same results in a much parsimonious way, while permitting the exploration of all model uncertainties. Our work offers a robust and more transparent approach to estimate one of the most important indicators guiding our policies on water security worldwide.

[1] Department of Ecology and Evolutionary Biology and High Meadows Environmental Institute, Princeton University, NJ, USA. [2] Centre for the Study of the Sciences and the Humanities (SVT), University of Bergen, Bergen, Norway. [3] Department of Decision Sciences and BIDSA, Boconi University, Milan, Italy. [4] School of the Built Environment, University of Reading, Reading, United Kingdom. [5] Open Evidence Research, Universitat Oberta de Catalunya (UOC), Barcelona, Spain. ✉email: apuy@princeton.edu

Irrigation agriculture is at the forefront of global food security. With the potential to provide crop yields more than two times as large as dryland agriculture[1,2], irrigation agriculture currently produces ~40% of all food consumed worldwide in just 20% of the total cultivated land[3]. Its capacity to maximise yields per unit of terrain is conditional upon the investment of high labour inputs per surface unit and the provision of a steady freshwater supply, which relaxes the dependency of crops on rainwater seasonality[4,5]. This allows for year-round harvests while reducing adverse impacts on crops from dry spells. Such features make irrigation agriculture a key resource to buffer population growth in our context of climate change.

The demand of water by irrigation has been constantly rising over the last decades[6]. In general, it is expected to increase even further in the coming years due to changes in precipitation patterns, higher temperatures and the expansion of irrigated areas to meet the projected boost in food demands[7–12]. Acquiring reliable estimates of irrigation water withdrawals is thus regarded as a first step towards a more informed management of global freshwater resources[13,14], ultimately endowing us with better tools to ensure food security without damaging the water system. At present, there are two main approaches to calculate global irrigation water withdrawals:

1. FAO's/Aquastat's approach. This is based on country surveys, questionnaires and censuses, literature reviews and coordination among relevant national and international agencies[15]. Its drawbacks include unreliability of data due to bureaucratic and political constraints[16], national interests[17,18], difficulties in homogenising water withdrawal data reported through different methods, and missing data points (not all countries provide information or the reported data does not pass Aquastat's quality check)[19].

2. Through Global Hydrological Models, Land Surface Models or Land Earth Systems Models[20]. Here we collectively refer to all these models as Global Models (GMs). They are spatially-distributed algorithms that simulate, among others, past, present and future hydrological processes on a global scale. Irrigation water withdrawals are generally computed at a specific time step and spatial resolution with sub-models on evapotranspiration processes, crop types, agrarian calendars, irrigation efficiencies, fertilisation, meteorological forcings and irrigated areas[7,21–24]. Some limitations of GMs are their high computational demands, poor calibration and a complex design that precludes a thorough assessment of output uncertainties[25,26].

These drawbacks are amplified by uncertainties in crop types, growing seasons, agrarian practices, irrigated areas and local soil and climatic conditions[27]. Global irrigation water withdrawal estimates are therefore highly sensitive to the selection of the FAO's or the GMs' approach, and even the choice of a specific GM is a source of bias[7]. The reliance on multi-model ensembles of GMs allows for the obtainment of probabilistic estimates, yet it exacerbates the computational, opacity and uncertainty-related problems mentioned above[28]. Such flaws limit the utility of global irrigation water withdrawal estimates in the policy realm, where stakeholders and non-experts alike should be able to swiftly replicate the results or, at least, understand the main assumptions upon which the analysis is based[29–31].

Here we show that global irrigation water withdrawals can simply be obtained as a function of irrigated areas. We submit eight GMs and two FAO-based datasets to uncertainty and sensitivity analysis methods and demonstrate that the variability of irrigation water withdrawals is mostly described by the extension of irrigation[19,23,24,32–37]. This paves the way to an easier, cheaper and more transparent estimation of global irrigation water demands and permits a systematic examination of all crucial uncertainties for water security. It also suggests that GMs can improve by better acknowledging the relevance of irrigated areas in their simulations. Our results align with recent works arguing that simple models may be more robust and of greater use than more elaborate approaches, especially when the estimation of interest is fraught with irreducible uncertainties[31,38].

## Results

**Irrigated areas and irrigation water withdrawals are strongly related**. Irrigated areas in GMs are parametrised with the Global Map of Irrigated Areas (FAO-GMIA)[39], a gridded product that documents the extension of irrigation at a 5 arcmin resolution. A linear trend between the areas reported by the FAO-GMIA and the irrigation water withdrawals simulated by GMs is apparent at the country level from 1900 up to 2005–2010, the last period for which there is systematic data available for both variables. This pattern holds regardless of the GM used (Figs. S1–S8). Here we focus on data from 2005 as it adequately summarises this historical relationship and facilitates comparison with two different FAO-based datasets, which reflect country-based irrigation water withdrawals in 2010–2012[19,24].

For most combinations of continent and irrigation water withdrawal dataset (except some particular cases for Europe, see Figs. S9–S11), the trend between irrigated areas and irrigation water withdrawals is well modelled by a linear regression in which irrigated areas are the predictor $x$ and water withdrawal is the response $y$ (Fig. 1). Such an approach fits well with previous works connecting these variables both empirically and theoretically[14,40]. Other parameters or intermediate outputs of GMs used to compute irrigation water withdrawals, such as irrigation efficiencies, total evapotranspiration or potential evaporation, do not appear to have any significant influence (Figs. S12–S14).

The strength of the relationship between irrigated areas and irrigation water withdrawals can be assessed with the coefficient of determination $r^2$, which measures how much variance in $y$ can be predicted from $x$. We check how $r^2$ changes when the main uncertainties conditioning its computation vary within reasonable bounds: for $c = 1, 2,..., m$ countries, we vary in a Monte-Carlo setting (see Methods):

- $X_1$: The selection of the GM or FAO-based dataset to characterise $y_c$.
- $X_2$: The multivariate method used to model a distribution for $y_c$ in case it is a missing value.
- $X_3$: The final sampled value from that distribution to impute $y_c$.
- $X_4$: The use of a robust or non-robust regression to estimate $r^2$, as some $y_c$ values are outliers (Fig. S11).

The coefficient of determination $r^2$ leans towards high values for Africa ($0.75 \le r^2 \le 0.9$, $P_{2.5}$, $P_{97.5}$), Asia ($0.68 \le r^2 \le 0.92$) and the Americas ($0.68 \le r^2 \le 0.95$) (Fig. 2a). The distribution of $r^2$ for these continents is clearly left skewed, with the smaller mode at approximately $r \le 0.8$ produced by the simulations conducted with just one or two GMs (CLM45 for Africa, CLM45 and MPI-HM for the Americas, and CLM45 and VIC for Asia) (Fig. 2b). The goodness of fit for Europe shows the largest spread ($0.5 \le r^2 \le 0.89$) and a three-modal distribution, with the highest $r^2$ values produced by MPI-HM and VIC and the lowest by PCR-GLOBWB.

For all continents, the most influential factor conditioning $r^2$ is the selection of the GM or FAO-based dataset to parametrise $y_c$ ($X_1$) (Fig. 2c). $X_1$ explains from 72% (Africa) to 95% (Asia) of the variance in $r^2$ values. In the case of the Americas, the use of a

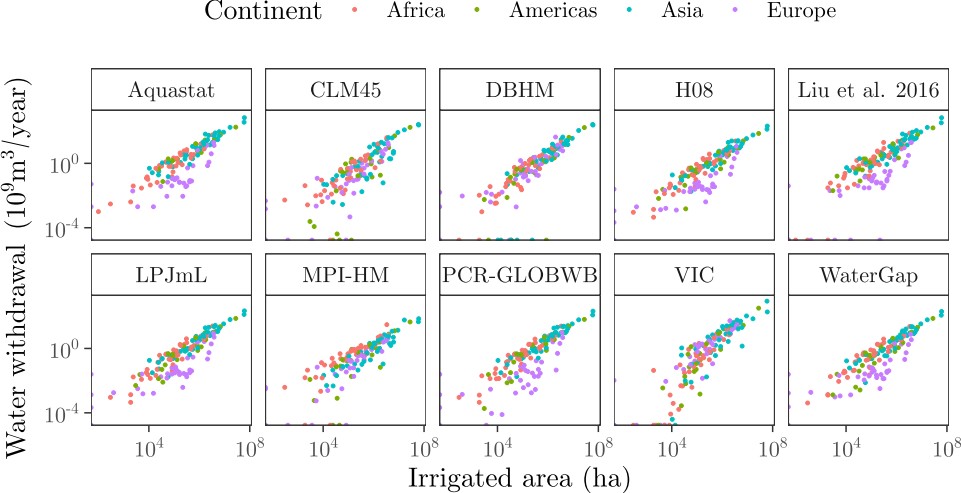

**Fig. 1 Scatterplots of irrigated areas against irrigation water withdrawals.** Irrigated area data are provided by the FAO-GMIA[37], which reflects the extension of irrigation in 2005. Irrigation water withdrawals are retrieved from GMs (CLM45, DBHM, H08, LPJmL, MPI-HM, PCR-GLOBWB, VIC, WaterGap) and from FAO-based datasets (Aquastat, Liu et al. [19]). GM data is from 2005 and FAO-based data from 2012. Each dot is a country. The abscissa is $x$ and the ordinate $y$ in Eq. (1) (see Methods).

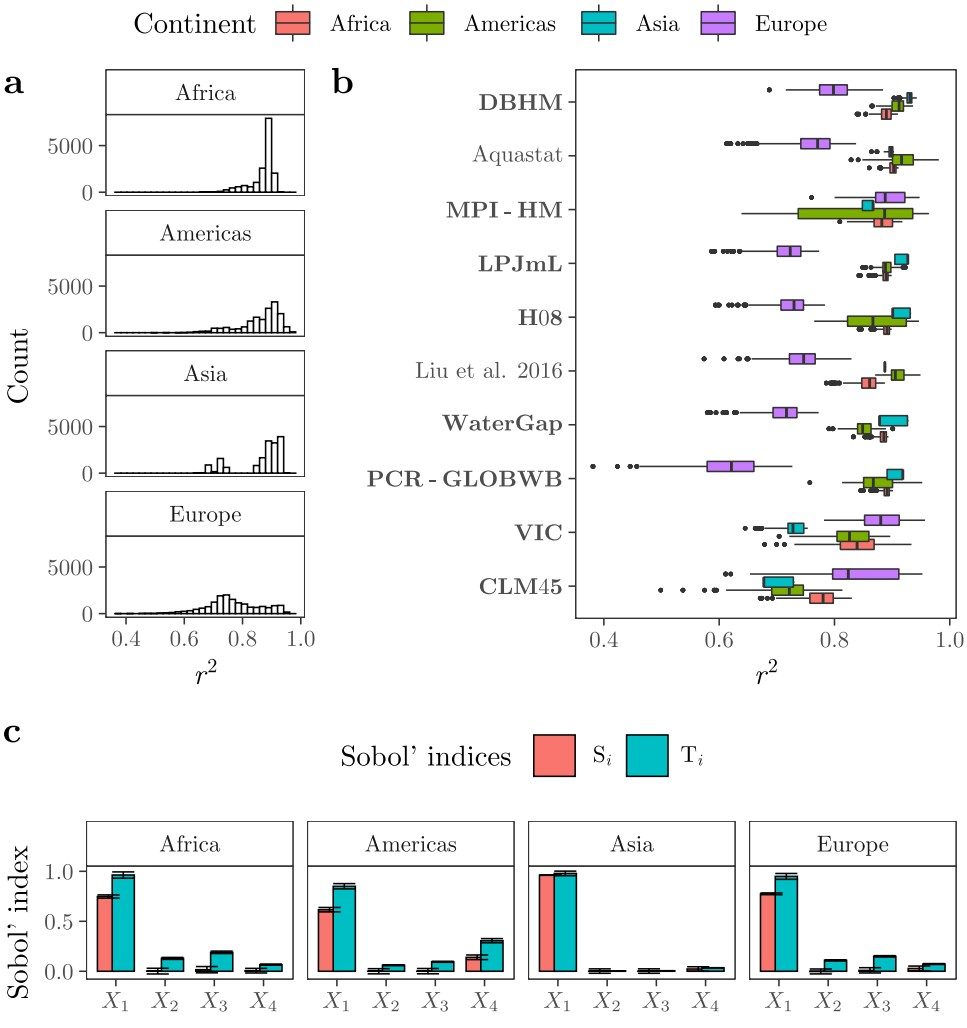

**Fig. 2 Uncertainty and sensitivity analysis. a** Empirical distribution of $r^2$ at the continental level. **b** Boxplots of $r^2$ values yielded by each GM (in bold) and FAO-based dataset (centre line, median; box limits, upper and lower quartiles; whiskers, 1.5x interquartile range; points, outliers). **c** Sobol' indices. $S_i$ and $T_i$ refer respectively to Sobol' first and total order indices. $S_i$ measures the influence of a parameter in the model output ($r^2$ in this case), while $T_i$ measures the influence of a parameter jointly with its interactions. The bars reflect the mean value and the error bars display the 95% confidence intervals, computed with the normal method after bootstrapping ($R = 1000$).

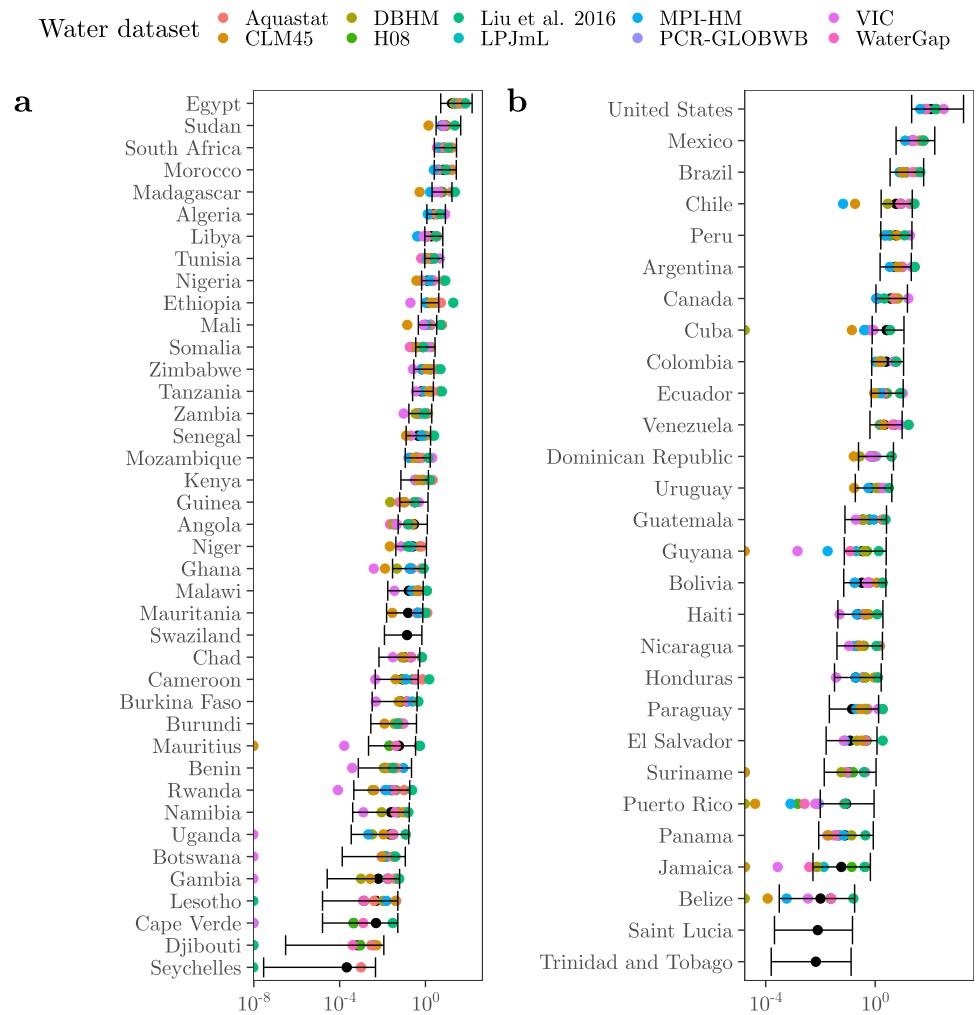

Irrigation water withdrawal $(10^9 \text{m}^3/\text{year})$

**Fig. 3 Comparison between our predictions and the GMs estimations.** The black dots and the error bars show the median, the minimum and the maximum irrigation water withdrawal values predicted from the irrigated areas reported by the FAO-GMIA[96] (Methods). The coloured dots show the irrigation water withdrawal estimates outputted by the GMs (DBHM, H08, LPJmL, MPI-HM, PCR-GLOBWB, VIC, CLM45, WaterGap) or reported by the FAO-based datasets (Aquastat, Liu et al.[19]). **a** Africa. **b** Americas.

robust or non-robust approach to compute $r^2$ ($X_4$) conveys an extra 12% of the uncertainty in the goodness of fit, with the robust option yielding slightly higher $r^2$ values on average (Fig. S15). The rest of the variance is due to second and third-order effects. For instance, the third-order effect between ($X_1, X_2, X_3$) conveys ~5%, ~10% and ~10% of the variance in $r^2$ for the Americas, Europe and Africa, respectively. An important fraction of the ambiguity in the goodness of fit is hence largely irreducible for it emerges as the joint effect of three different structural uncertainties (Fig. S16).

**Estimating irrigation water withdrawals from irrigated areas.** Such linear relation and high $r^2$ values suggest that irrigated areas might fairly predict irrigation water withdrawals, especially for countries in Africa, Asia and the Americas. We thus combine Eq. (1) with an uncertainty analysis to estimate irrigation water withdrawals as a function of irrigated areas and compare the predictions with the ten point estimates yielded by all GMs and FAO-based datasets considered (see Methods).

The results show that GMs and FAO-based estimates fall nicely within the ranges defined by our predictions for a very large majority of countries (Figs. 3 and 4). Ninety-nine countries out of 139 (71%) present seven or more estimations bounded by the error bars of our regressions, while 26 countries (18%) have all ten point estimates fully framed. Examples of the latter are Egypt, South Africa, the United States, Mexico, Brazil, Afghanistan, India, Pakistan, Italy, Spain or France, all of them top-ranking countries in irrigation water consumption. For China, also a major consumer of irrigation water, our approach frames nine out of ten point estimates, with VIC falling outside our interval. Figures 3 and 4 thus offer a realistic impression of the uncertainty associated to these predictions.

The number of point estimates framed by our predictions increases with the number of countries in all four continents (Fig. S17). Malta is the only country for which our ranges do not embed any previous estimate. Other countries for which our predictions fit the preexisting estimates poorly are Seychelles (only one estimation framed), Cuba and Kuwait (2), or Ethiopia, Puerto Rico, Indonesia and the Philippines (3). All these

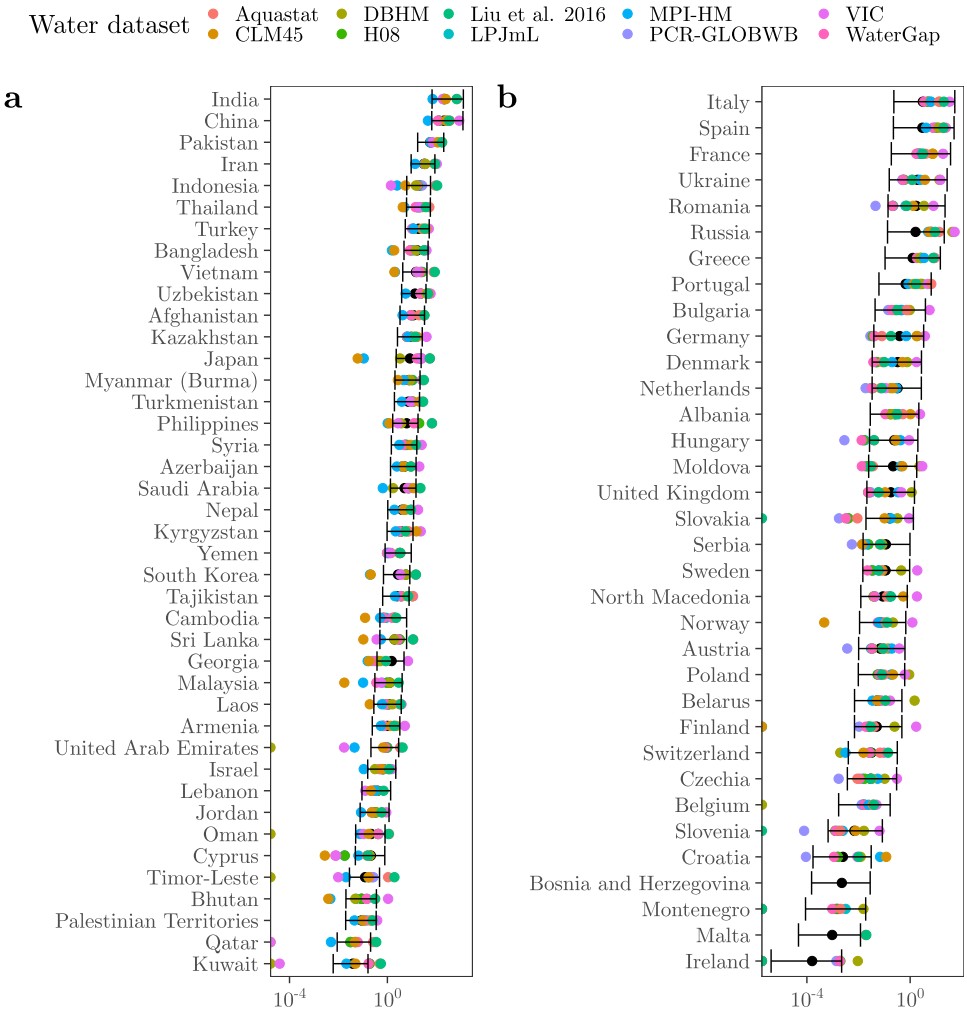

**Fig. 4 Comparison between our predictions and the GMs estimations.** The black dots and the error bars show the median, the minimum and the maximum irrigation water withdrawal values predicted from the irrigated areas reported by the FAO-GMIA[96] (Methods). The coloured dots show the irrigation water withdrawal estimates outputted by the GMs (DBHM, H08, LPJmL, MPI-HM, PCR-GLOBWB, VIC, CLM45, WaterGap) or reported by the FAO-based datasets (Aquastat, Liu et al.[19]). **a** Asia. **b** Europe.

countries share a large uncertainty with regard to the irrigation water withdrawal estimates produced by FAO and GMs. The datasets that show the most point estimates beyond or above our error bars are VIC in the case of Africa (~51%), Asia (~58%) and Europe (~43%), and CLM45 (~33%) in the case of the Americas (Fig. S18).

Our approach also replicates the irrigation water withdrawal estimates produced by GMs for 2050 in a context of climate change, regardless of the social context or the Representative Concentration Pathway (RCP) selected (Figs. S19 and S20, RCP2.6, RCP6, RCP8.5; see Methods). For most countries, our ranges encompass the estimates that GMs simulate for the future as nicely as those they yield for the present. This is shown in Fig. 5, with a large majority of countries clustering on the upper right side of the plot. This area includes 80% of the countries and contains the largest agricultural water consumers (e.g. China, India, Spain, Italy, Egypt or the United States, among others). In the case of Kuwait, Moldova or Angola, our approach mimics future estimates better than current estimates, while the opposite is true for Chile, Croatia or Finland.

**The relation may scale down.** The trend between irrigated areas and irrigation water withdrawals detected at the country level also emerges at smaller geographical scales. To illustrate this phenomenon, we show independent data at three levels: (1) at the irrigation system level, from the Australian National Committee on Irrigation and Drainage (ANCID)[41,42]; (2) for every county of Colorado (a state whose farm water use exceeds USA's national average[43]), from the Colorado Water Conservation Board[44]; and (3) for every state of the USA, from the US Department of Interior[45]. In all three cases, the proportion of the variability in irrigation water withdrawals that is described by irrigated areas (circa 0.7–0.9, 95% CI) is very similar to the results obtained at the national level with the FAO-based and GMs datasets (Fig. 6).

Irrigated areas drive water withdrawals even at the grid cell level, the minimum geographical unit in which GMs simulate irrigation water withdrawals. This is especially the case with CLM45 and MPI-HM, which operate like a linear model despite their computational complexity[46], pp. 346–365 (Figs. S23–S54). The same can be said regardless of the GM for the cells of Egypt, Morocco, Sudan, South Africa and Zimbabwe in Africa; of most

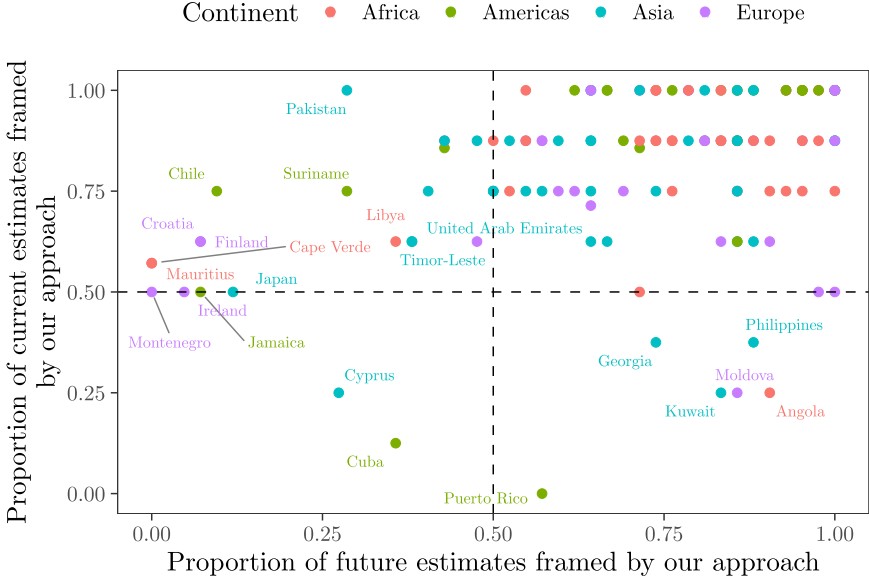

**Fig. 5 Proportion of current and future irrigation water withdrawal values yielded by GMs that are framed by our predictions.** The proportion in the *y* axis is taken over eight GMs (e.g., a dot closer to *y* = 1 means that our range of predictions covers all eight estimates yielded by GMs), whereas in the *x* axis it is taken over four GMs running on different combinations of the Shared Socioeconomic Pathway 2 (SSP2) and RCPs (e.g., two combinations for PCR-GLOBWB, five for LPJmL, five for H08, two for MPI-HM). See Puy[95], p. 45 for a full specification of the combinations. A dot closer to *x* = 1 means that our range of predictions covers all these 14 estimates. Only the countries showing <50% in either the *x* or the *y* axis are labelled.

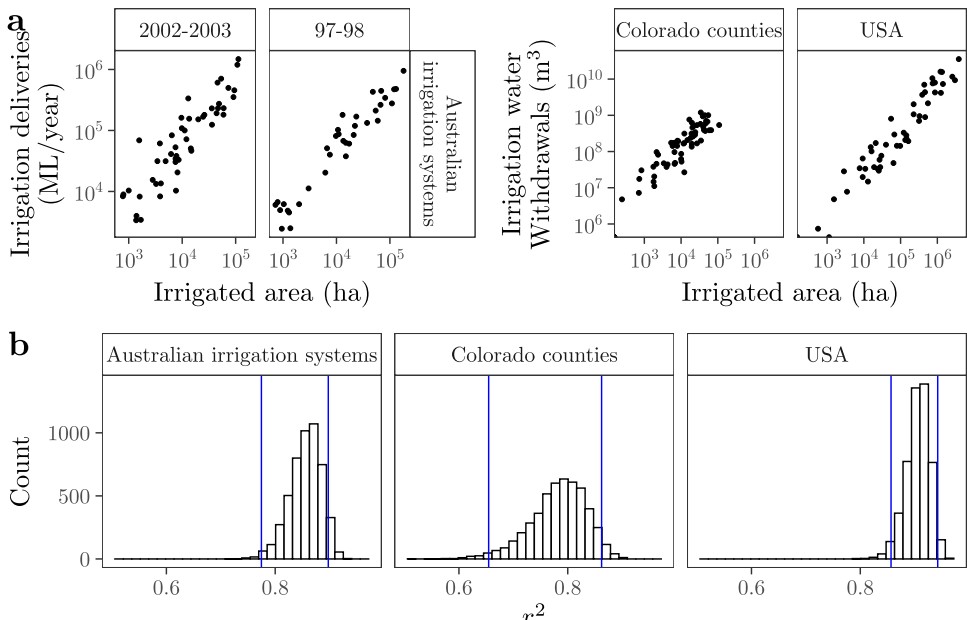

**Fig. 6 Examples of the relation between irrigated areas and irrigation water withdrawals at smaller geographical scales. a** Scatterplots presenting data for *circa* 60 Australian irrigation systems[41,42], Colorado counties[44] and North-American States[45]. **b** Distribution of $r^2$ after bootstrapping (*R* = 5000). The blue lines show the 95% confidence intervals, calculated with the bias-corrected and accelerated (bca) method[97]. See Figs. S21 and S22 for the regression diagnostics.

countries in Asia (including China and India); of Mexico, the USA, Colombia, Argentina or Peru in the Americas; and of Spain, France, Italy and Russia in Europe. Given that irrigated water withdrawals at the grid cell level are often aggregated to produce estimates at the river basin or at the agro-ecological level[23,47], irrigated areas may also drive irrigation water withdrawals at these scales in the countries and GMs just mentioned.

**How do irrigation water withdrawals scale with irrigated areas?** The tight relation between irrigated areas and irrigation water

withdrawals enables the assessment of how the latter responds to changes in the former. This is expressed by $\beta$, the slope of the linear regression of $\log(y)$ against $\log(x)$ (see Methods, Eq. (1)). If $\beta < 1$ ($\beta > 1$), every increase in the extension of irrigation leads to marginal (accelerated) increases in irrigation water consumption. This framework is known as scaling[48,49], and allows to explore (1) whether larger irrigated areas are, on average, less water efficient than smaller ones ($\beta > 1$), and (2) whether the complexity behind irrigation water withdrawals can be further simplified to a single $\beta$ value.

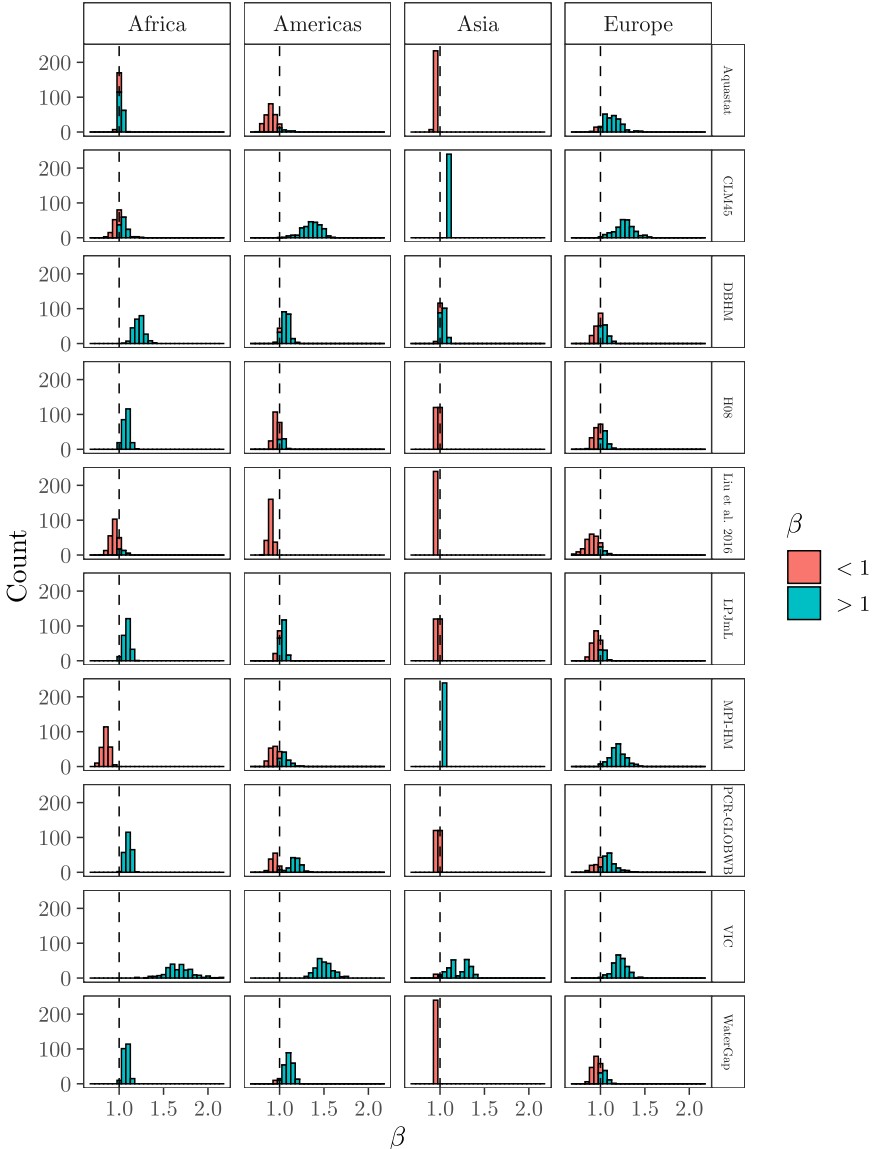

**Fig. 7 Distribution of $\beta$ after accounting for all the uncertainties in the computation of Eq. (1) ($X_1$, $X_2$, $X_3$, $X_4$).** Each distribution is formed by a population of 240 $\beta$ values. The histogram in the Liu et al.[16] facet for Asia consists of 17 different yet very similar $\beta$ values and hence its single bin. The vertical, dashed line is at $\beta = 1$ (see Methods).

At the continental level, $\beta > 1$, $\beta \approx 1$ or $\beta < 1$ depending on the GM or FAO-based dataset selected to characterise $y_c$. This is exemplified by Africa, which shows $\beta > 1$ under DBHM, H08, LPJmL and PCR-GLBWB, VIC and WaterGap, $\beta < 1$ under MPI-HM, and $\beta \approx 1$ under Liu et al.[19] and CLM45 (Fig. 7). Such volatility originates from the uncertainty in $y_c$ and currently prevents from inferring the existence of a consistent scaling relationship between both variables. For Australian irrigation systems and Colorado counties, $\beta$ is indistinguishable from 1, indicating that irrigation water withdrawals tend to become twice as large if irrigated areas at the system or the county level are doubled in size. This contrasts with the USA, whose $\beta > 1$ suggests that states with a larger extension of irrigation have a disproportionate consumption of irrigation water given the size of their irrigated areas (Fig. S55).

## Discussion
The present paper shows that irrigated areas describe a large degree of variability in irrigation water withdrawals, and that the latter can be approximated as a function of the former. These results are grounded on annual irrigation water withdrawal estimates outputted at the national level by eight Global Models (GMs) and reported by two FAO-based datasets. They are also based on independent data retrieved at the state, county and irrigation system level. Hence there is a great potential for the simplification of methods to calculate irrigation water withdrawals, especially with regard to those employed at larger geographical scales.

That a complex variable such as the volume of water withdrawn for irrigation is nicely described by just a single factor may appear surprising given the large space of relevant factors influencing its behaviour (e.g., crop type and calendars, growing seasons, irrigation efficiency, climate, crop evapotranspiration, soil texture). Yet several degrees of freedom are often determined or summarised by a small set of constraints or even by a single parameter. Size is one of such parameters: for animals, it defines their strength, metabolic rate, life span or population density[50,51]; for cities, its pace of innovation, number of patents or total

electrical consumption[48,52]. The size of irrigated areas (their extension) appears to be a similar driving force for irrigation water withdrawals.

The irrigation module of GMs contains parameters or secondary outputs whose influence in the calculation of water withdrawals is very minor or inconsequential [e.g., total evapotranspiration, potential evaporation or irrigation efficiency once controlled for irrigated area, Figs. S12–S14]. The effect of parameters such as crop coefficients or crop calendars, which vary as a function of time, or of complex sub-models such as fertilisation schemes or the climate driver[33–35,53], is much harder to check. However, if a linear regression can nicely fit the input-output mapping of complex irrigation algorithms, other similar fast-running statistical emulators may be able to work in the computationally expensive sub-models nested within GMs. Substituting these sub-models with time-effective emulators can be an effective way to save computational resources and bridge model realism with computational efficiency. This may allow modellers to focus on answering relevant water policy questions without the extra burden of managing unneeded complexity.

The strong influence of irrigated areas makes GMs very sensitive to the FAO-GMIA[39], the gridded map used by GMs to parametrise global irrigated areas. Yet the FAO-GMIA is just one of the five datasets currently available on the extension of irrigation[18,54–57]. Depending on the dataset selected, the irrigated area of a given country can differ by up to four orders of magnitude[8], a range that reflects our limited knowledge on the current size of irrigated agriculture. The FAO-GMIA was the only map available when most GMs were initially designed, but research conducted over the last ten years has broadened the range of products available and increased the uncertainty range[18,56,57]. By relying exclusively on the FAO-GMIA, GMs discount this source of ambiguity and yield estimates that are critically conditioned by a structural model design.

Let us illustrate this issue with the case of China and India, for instance. According to the FAO-GMIA, their irrigated areas extend over 61 Mha. This point estimate turns into ranges that respectively span 43–74 Mha and 15–88 Mha if the other irrigated area datasets are taken into account[18,56,57]. These divergences are explained by the different methodological approaches mobilised to map irrigated areas, including the definition of what is considered an "irrigated area" and the degree of reliance on official statistics. Given the strong weight of the extension of irrigation, the already large variance in irrigation water withdrawals displayed in Figs. 3 and 4 will become much larger if GMs factor this uncertainty in. The same will apply to the estimates of future irrigation water withdrawals, because the uncertainty of global irrigated areas in 2050 spans half an order of magnitude (300–800 Mha, with the most extreme values reaching 1800 Mha)[8].

In light of these results, the addition of conceptual depth aiming at making GMs more accurate (by modelling the human influence, increasing the spatial resolution or running multi-model ensembles) appears questionable[20,58]. The fastest way to acquire more precise irrigation water withdrawal estimates seems to be through a better appraisal of irrigated areas. This may also provide sharper insights into the growth rate between irrigated areas and water demands (i.e. whether $\beta < 1$, $\beta \approx 1$, $\beta > 1$, see Fig. 7). Scaling relationships might hold crucial information for discerning the role of size in the sustainability of irrigated agriculture[40,59–61], as well as for our design of irrigation schemes. Trivially, given 1000 ha potentially irrigable, should we promote one system extending over the whole 1000 ha or 10 systems of 100 ha each? If size largely drives some properties of irrigated agriculture, it may be that excess complexity delays—rather than

accelerates—our understanding of irrigation systems, thus posing a hindrance to the design of robust water policy responses. This observation resonates with a broad literature showing that too much detail in model efforts undercuts reliable management[62–64], as well as with recent warnings against excess complexity in mathematical modelling[65].

That GMs are too complex given the quality of the data available is also suggested by the ambiguity surrounding other aspects of their irrigation module. Model inputs such as the crop coefficient or the evapotranspiration equation, among others, are also uncertain[66–71]. Their effect in the model output is likely to be minor given the strong weight of irrigated areas, yet we can not rule out their being influential through interactions. The computation of irrigation water withdrawals in GMs relies on multiplications, divisions and exponentials (e.g., Eq. (1) in Wada et al.[32]; Eqs. (1) and (2) in Döll and Siebert[14], Eqs. (7)–(11), A1–A3 in Jägermeyr et al.[34]). Such operations promote non-additivities, whose effect on irrigation water withdrawals can only be appraised with a global sensitivity analysis (GSA), i.e. by moving all uncertainties at once. However, the literature on GMs has relegated GSA in favour of ensemble, one-at-a-time or piecewise sensitivity analysis[7,72,73]. These techniques are computationally affordable but severely underpowered to scrutinise the input space, and are unable to detect interactions[74,75].

The much parsimonious approach adopted here has a predictive power almost identical to that of FAO's and eight GMs combined. It has empirical support at several geographical scales, allows to save personal, financial and computational resources, and facilitates an appraisal of uncertainties and sensitivities, including those related with irrigated areas. An extension of our work is to explore why the trend between irrigated areas and water withdrawals is the weakest in European countries. Another direction is to assess whether irrigated areas drive irrigation water withdrawals in other irrigation systems and/or in different sub-regional contexts. This will help assess the extent to which this trend scales down in a robust way.

Finally, we should stress that we are not proposing to substitute approaches relying on crop, soil and climate parameters for linear regressions of irrigated areas. At small granularities, for instance at the plot or the scheme level, these methods nicely appraise physical process and facilitate monitoring of water withdrawals through time. What we argue is that so much detail may not be warranted at larger scales or under strong uncertainties. The debate between proponents of computationally-intensive methods and advocates of more simple approaches is vibrant in the climate modelling community[64,76]. Both parties also need to be heard in the field of global hydrology.

## Methods

**Data collection**. We use irrigation water withdrawal values outputted by eight GMs [six Global Hydrological Models (PCR-GLOBWB[32,53], H08[33], LPJmL[34], WaterGap[35], MPI-HM[22], DBHM[23]), one Land Surface Model (VIC[77]) and one Land Earth System Model (CLM45[46])]. For PCR-GLOBWB, H08, LPJmL and WaterGap we rely on the products generated by Huang et al.[21], who downscaled the data yielded by these four GMs between 1971-2010 with Aquastat water withdrawal estimates. For DBHM, MPI-HM, VIC and CLM45 we use the data produced by the Inter-Sectoral Impact Model Inter-comparison Project (ISI-MIP)[78]. Our analysis focuses on the values reported for 2005 in all the cases. All GMs datasets used are forced by WFDEI climate data except MPI-HM and CLM45, which are forced by MIROC5 and GFDL respectively.

We also retrieve irrigation water withdrawal data from two FAO-based datasets. We use the Aquastat country-level data produced by Frenken and Gillet[24] for 2012 and the dataset elaborated by Liu et al.[19], who filled out missing values in the Aquastat dataset using inverse distance weighting, nearest neighbour or linear interpolation based on associated variables.

For irrigated areas, we collect the data generated by the FAO-GMIA at the national level from Meier et al.[57], which documents the extension of irrigation at c.

2005. In order to assess the influence of the FAO-GMIA at the cell level, we retrieve the data produced by the Historical, gridded land use (HYDE 3.2) product from ISI-MIP[78]. The HYDE 3.2 relies on the FAO-GMIA and the MIRCA 2000 to parametrise irrigated areas[79], with the MIRCA 2000 being also a gridded product grounded on the FAO-GMIA[80], p. 5).

To investigate whether our approach replicates the irrigation water withdrawal estimates produced by GMs for 2050, we retrieve from ISI-MIP the data produced by PCR-GLOBWBW, LPJmL, H08 and MPI-HM under five different social, climatic, and $CO_2$ scenarios[81]:

- rcp26/rcp26: Water abstraction and land use (including irrigated areas) change according to the Shared Socioeconomic Pathway 2 (SSP2, "Middle of the road"). Future climate and $CO_2$ concentration evolve as outlined by the Representative Concentration Pathway 2.6 (RCP2.6, mean temperature increase of 1°C up to 2065, $CO_2$ emissions declining by 2020).
- rcp60/rcp60: Water abstraction and land use (including irrigated areas) change according to SSP2. Future climate and $CO_2$ concentration evolve as outlined by the Representative Concentration Pathway 6.0 (RCP6.0, increase of 1.4 °C, $CO_2$ declining by 2080).
- 2005soc/rcp26: Land use (including irrigated areas), nitrogen deposition and fertilizer input are fixed at 2005 values. Future climate and $CO_2$ concentration as in RCP2.6.
- 2005soc/rcp60: Land use (including irrigated areas), nitrogen deposition and fertilizer input are fixed at 2005 values. Future climate and $CO_2$ concentration as in RCP6.0.
- 2005soc/rcp85: Land use (including irrigated areas), nitrogen deposition and fertilizer input are fixed at 2005 values. Future climate and $CO_2$ concentration as in RCP8.5.

**Data treatment**. The GMs just mentioned have a spatial resolution of 0.5° × 0.5° and compute irrigation water withdrawals in each cell at a monthly time step. For each GM, we retrieve the data from 2005 and allocate each cell to a specific country given its geospatial information (longitude and latitude). We produce annual irrigation water withdrawal values at the national level by adding the values of all cells within the same country. We then bind all GMs datasets with the Aquastat and the Liu et al.[19] datasets and pair each country with the national irrigated areas reported by the FAO-GMIA. This procedure yields missing values in water withdrawal for 28 unique countries, a total of 69 missing data points (Table S1, Fig. S56).

To calculate the relation between irrigated areas and water withdrawal at the cell level, we merge the HYDE 3.2 with all the GMs and pair only the cells that show the same coordinates in both products.

**The model**. Following the linear trend between irrigated area and irrigation water withdrawals at the country level (Fig. 1), we model their relation as

$$\log(y_c) = \alpha + \beta \log(x_c) , \qquad (1)$$

where $y_c$ and $x_c$ are respectively the irrigation water withdrawal and the irrigated area of country $c$, for $c = 1, 2, ..., m$ countries. $\alpha$ is a constant and $\beta$ the scaling exponent describing the growth rate between $x_c$ and $y_c$.

**Uncertainty analysis**. There are four main sources of uncertainty that condition the goodness of fit of Eq. (1), estimated with the $r^2$ value. We treat these uncertainties as triggers ($X_1, X_2, X_3, X_4$), i.e. random parameters that explore the uncertainty in the model design space. They are the following:

- $X_1$: The selection of the GM or FAO-based dataset to characterise irrigation water withdrawals at the country level ($y_c$ in Eq. (1)). There are ten different alternatives (eight GMs and two FAO-based datasets, see Fig. 1).
- $X_2$: The multiple imputation methods used to impute missing values. After pairing the eight GMs and two FAO-based datasets with the irrigated areas reported by the FAO-GMIA, some countries showed missing irrigation water withdrawal values. To ensure that $x$ and $y$ have the same individual data points across all data sets, we replace missing values with substituted values using multiple imputation methods. Unlike single imputation, which treats the imputed value as the "true" value, multiple imputation accounts for the uncertainty about the prediction of the missing value by randomly drawing $d$ values from a distribution specifically modelled for each missing entry[82]. This creates $d$ different completed datasets or imputations. Given the linear trend observed in Fig. 1, we assess how three different regression-based, multiple imputation methods affect the estimation of $y_c$: Bayesian regression, linear regression ignoring the model error, and linear regression with bootstrap. The Bayesian regression method imputes $y_c$ by the normal model defined by Rubin[83], while the linear regression with bootstrap method draws a bootstrap sample from $x$ and $y$, calculates regression weights and imputes with normal residuals[84].
- $X_3$: The selection of the completed dataset to compute $r^2$. The number of imputations $d$ to obtain an appropriate estimation of the true missing value has long been a topic of discussion. Graham et al.[85] recommend 20

imputations for 20–30% missing data and 40 imputations for 50% missing data. The number of missing data points in our study is smaller than 10% for almost all continents and datasets, except for Aquastat in the Americas (c. 40%) (Fig. S56). In order to ensure enough statistical power, we set the number of imputations at $d = 40$ and create 40 different completed datasets in each iteration.

- $X_4$: The eventual use of corrective measures to calculate the line of best fit in case $y_c$ is an outlier. Outliers can bias the estimation of $r^2$. We document their presence for some continents depending on the irrigation water withdrawal dataset used (Fig. S11). The classic estimator of $r^2$ when there is an intercept term in the linear model is

$$r^2 = \left( \frac{\sum_{c=1}^{m}(y_c - \bar{y})(\hat{y}_c - \bar{\hat{y}})}{\sqrt{\sum_{c=1}^{m}(y_c - \bar{y})^2 \sum_{c=1}^{m}(\hat{y}_c - \bar{\hat{y}})^2}} \right)^2 , \qquad (2)$$

where $y_c$ is the observed irrigation water withdrawal value for the country $c$, $\bar{y}$ the mean, $\hat{y}$ the fitted value and $\bar{\hat{y}}$ the mean predicted responses. In order to account for the effect of applying corrective measures to outliers, we consider the consistency-corrected formula by Renaud and Victoria-Feser[86],

$$r^2 = \frac{\sum_{c=1}^{m} w_c \left( \hat{y}_c - \bar{\hat{y}}_w \right)^2}{\sum_{c=1}^{m} w_c \left( \hat{y}_c - \bar{\hat{y}}_w \right)^2 + a \sum_{c=1}^{m} w_c \left( y_c - \hat{y}_c \right)^2} , \qquad (3)$$

where $\bar{\hat{y}}_w = (1/\sum w_c) \sum w_c \hat{y}_c$, $a$ is a correction factor set at 1.2 and the weights $w_c$ and the predicted values $y_c$ are produced by the fast S-algorithm of Salibian-Barrera and Yohai[87].

To assess how $X_1 ..., X_4$ condition the final $r^2$ value, we conduct a Monte–Carlo-based uncertainty analysis. We design a $(N, 2k)\mathbf{Q}$ sample matrix using Sobol' Quasi-Random Numbers[88,89]. The Sobol' sequence is a base-2 sequence that explores the uncertainty space more effectively than random numbers, for it leaves smaller unexplored volumes. After a few experiments we decided to set the number of rows at $N = 2^{13}$ to handle a sample size large enough to ensure the convergence of the Sobol' indices (see section "Sensitivity analysis" below).

We allocate the leftmost $k$ columns of $\mathbf{Q}$ to an $\mathbf{A}$ matrix and the rightmost $k$ columns to a $\mathbf{B}$ matrix. In these matrices each row is a sample point and each column a trigger described with a probability distribution according to its uncertainty (Fig. S57). Any point in either $\mathbf{A}$ or $\mathbf{B}$ can be referred to as $x_{vi}$, where $v$ and $i$ respectively index the row (from 1 to $N$) and the column (from 1 to $k$). We also create $k\mathbf{A}_B^{(i)}$ matrices, where all the columns come from the $\mathbf{A}$ matrix except the $i$-th, which comes from the $\mathbf{B}$ matrix (Fig. S58). The $\mathbf{A}_B^{(i)}$ matrices are required to compute the Sobol' indices of the triggers (see section "Sensitivity analysis" below)[90]. Overall, this design has a computational cost $C$ of $C = N(k+2) = 2^{13}$ $(4 + 2) = 49,152$ model runs per continent.

Our algorithm runs rowwise, as follows: for $v = 1, 2, ..., C$ rows, it selects the irrigation water withdrawal dataset according to $X_{1_v}$, fills the missing values in $y_c$ given the conditions set by $X_{2_v}$ and $X_{3_v}$, and finally computes Eq. (1) + Eq. (2) or Eq. (1) + Eq. (3) depending on the criteria defined by $X_{4_v}$. The model output in the $v$-th row is therefore a specific $r_v^2$ value calculated according to the conditions established by $X_{1_v}, ..., X_{4_v}$. To obtain the range of predictions shown in Figs. 3 and 4, we retrieve $x_c$ from the FAO-GMIA and compute Eq. (1) with the 2,400 paired $\alpha_v$ and $\beta_v$ coefficients obtained from the simulations.

**Sensitivity analysis**. We conduct a global sensitivity analysis using Sobol' indices[91,92], which decompose the variance of the model output $V(y)$ into fractions that are attributed to the model inputs, as

$$V(y) = \sum_{i=1}^{k} V_i + \sum_{i}\sum_{i<j} V_{ij} + ... + V_{1,2,...,k} , \qquad (4)$$

where

$$\begin{aligned} V_i = V_{x_i}\left[ E_{\mathbf{x}_{\sim i}}(y|x_i) \right] \quad V_{ij} &= V_{x_i x_j}\left[ E_{\mathbf{x}_{\sim ij}}(y|x_i, x_j) \right] \\ &- V_{x_i}\left[ E_{\mathbf{x}_{\sim i}}(y|x_i) \right] \\ &- V_{x_j}\left[ E_{\mathbf{x}_{\sim j}}(y|x_j) \right] \end{aligned} \qquad (5)$$

and so on up to the $k$th order. $V_i$ is the conditional variance of $x_i$ on $V(y)$, $V_{ij}$ the conditional variance of $x_i$ and $x_j$ on $V(y)$, etc. The notation $E_{\mathbf{x}_{\sim i}}(y|x_i)$ means that the mean $y$ value, represented by the $E(.)$ operator, is taken over all inputs except $x_i$. Sobol' indices are then calculated as

$$S_i = \frac{V_i}{V(y)} \quad S_{ij} = \frac{V_{ij}}{V(y)} . \qquad (6)$$

$S_i$ represents the first-order effect of $x_i$; $S_{ij}$ is the second-order effect of $(x_i, x_j)$ (formed by the first-order effect of $x_i$ and $x_j$ and their interaction), etc. $S_i, S_{ij} ...$ can be interpreted as the reduction in variance that will be obtained in the model output if $x_i, (x_i, x_j),...$ are fixed to their "true value", i.e., if they are no longer

uncertain. These reductions are of course averaged over all the possible values of the unknown "true" value.

We also calculate the total-order index $T_i$, which assesses the first-order effect of a model input jointly with its interactions[92]. When $T_i > S_i$, $x_i$ is involved in interactions. $T_i$ is calculated as

$$T_i = 1 - \frac{V_{\boldsymbol{x}_{\sim i}}\left[E_{x_i}(y|\boldsymbol{x}_{\sim i})\right]}{V(y)} = \frac{E_{\boldsymbol{x}_{\sim i}}\left[V_{x_i}(y|\boldsymbol{x}_{\sim i})\right]}{V(y)} \ . \tag{7}$$

There are several estimators available to compute Eqs. (6) and (7). Here we use the Jansen[93] estimators, considered best practice in sensitivity analysis[74,94]. The Jansen estimators make use of the model output $y$ produced after running the model $f$ in the $v$th row of the $\boldsymbol{A}$, $\boldsymbol{B}$ and $\boldsymbol{A}_B^{(i)}$ matrices. This is indicated as $f(\boldsymbol{A})_v$, $f(\boldsymbol{B})_v$ and $f(\boldsymbol{A}_B^{(i)})_v$:

$$S_i = \frac{V(y) - \frac{1}{2N}\sum_{v=1}^{N}\left[f(\boldsymbol{B})_v - f(\boldsymbol{A}_B^{(i)})_v\right]^2}{V(y)} \ , \tag{8}$$

$$T_i = \frac{\frac{1}{2N}\sum_{v=1}^{N}\left[f(\boldsymbol{A})_v - f(\boldsymbol{A}_B^{(i)})_v\right]^2}{V(y)} \ . \tag{9}$$

## Data availability

The irrigation water withdrawal data generated in this study, as well as the datasets needed to reproduce our results, are available in Puy[95] and in https://github.com/arnaldpuy/achilles_heel. The irrigation water withdrawal estimates produced by GM can be retrieved in https://www.isimip.org.

## Code availability

The R code to replicate our results is available in Puy[95] and in https://github.com/arnaldpuy/achilles_heel.

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

## Acknowledgements

Thanks to Francesc C. Conesa (Institut Català d'Arqueologia Clàssica) and to Jonas Meier (German Aerospace Center) for helping with the conversion of .asc files to .csv files. This work has been funded by the European Commission (Marie Skłodowska-Curie Global Fellowship, grant number 792178 to AP) and by the National Science Foundation grant DMS 1951358.

## Author contributions

A.P. designed the research, retrieved the data and conducted the simulations. A.P., E.B., S.L.P., S.A.L. and A.S. interpreted and discussed the results. A.P. lead the writing of the manuscript, with contributions from E.B., S.L.P., S.A.L. and A.S. All authors edited, revised and approved the final version.

## Competing interests

The authors declare no competing interests.
