## [Peer Review File · Nature Communications]

REVIEWER COMMENTS

Reviewer #1 (Remarks to the Author):

The manuscript "Irrigated areas drive irrigation water withdrawals" aims at assessing the efficacy of irrigated areas data to estimate the agricultural water use at continental scale using statistical approach. This work is novel and it has great potential to influence the conventional thinking in global information systems on Ag water. The methodology employed by the authors using Aquastat, FAO-GMIA and multiple hydrological models to calculate the gridded water withdrawals, in general, seems sound and interesting. I found this work impressive, and would definitely like to admire the efforts the authors put together for collecting, arranging, and analyzing the data. Though the sophisticated models for precise estimation of water use at large scale are inevitable, statistical approaches could provide a great advantages in terms of time and computational expense. Also, it is more convenient to communicate with the policy makers when moderately complex estimation techniques are used that are linked with widely used datasets e.g., FAO-GMIA. The manuscripts is well-written and flow of information is smooth; however, some improvements are needed before it can be recommended for publication. I would suggest authors to submit the revised version of the manuscript so it can be further reviewed. Following are my comments and suggestions for possible improvements:

Introduction

Please elaborate the purpose of this study in a separate paragraph. You have mentioned the two approaches for calculating global irrigation water withdrawals and described some of their shortcomings. It will be interesting to see what kind of gaps other researchers have highlighted related to irrigation water withdrawals estimations at global scale. You can link you literature review with the challenges in policy making based on robust knowledge, and substantiate the need of establishing less-complex and time-saving methods to evaluate the water use for irrigating crops.

Results

The results section is well elaborated and provides a good insight into the correlation of irrigated areas and water withdrawals. Authors included all major countries that practice irrigation worldwide but I was not able to find China in the text, though it is shown in figure 4a. Authors are encouraged to expand their findings from China and India as well since they are world's two largest countries with irrigation infrastructure. As authors might have seen in the literature that in China and India there are higher uncertainties in irrigated area reporting, and it would be suitable to provide more details in the results.

Line 143-154: Authors are explaining the interlinkage of irrigation water withdrawal values and the fraction of irrigated area in a cell. This is an important component of the results section and needs more attention. Therefore, I would suggest that authors add details on how the fraction is estimated in models and what systematic errors or discrepancies are associated with it. Also, if the models incorporate FAO-GMIA's gridded fraction for irrigated land, it should be clearly mentioned.

Line 155-172: Authors have mentioned that the slope of the linear regression of $\log(y)$ and $\log(x)$, denoted as β , which may be governed by the size of the irrigation schemes and complexity for estimating water withdrawals. I suggest authors to explore more factors that might affect this scaling framework. Only linking the β with the size of the irrigation scheme and calculation complexity may not be adequate from scientific point of view. Water productivity of the irrigated agriculture could be another factor. Furthermore, water laws of a country or state could play a role. Also, authors need to consider the fact that large irrigation areas may not have disproportionately higher evaporation, for example, large center pivot irrigated areas.

Discussion

Line 205-212: Do authors mean statistical emulators here? If that's the case, then it should be clearly stated. Authors should realize that it could raise further questions on the reliability of the mechanistic models if sub-models are replaced by the simple algorithms. Despite the fact that complexity leads to extensive computation, it still allows the modelers to keep a track of variables that have higher influence on the outputs. Though this study argues about the efficacy of the statistical methods that are time-saving, the advancement in modeling and benefits of assimilation of complex models with satellite based datasets cannot be ignored. Therefore, adding conceptual depths in models cannot be labeled as dubious pursuit just by skewing the discussion towards uncertainty in irrigated area data. I would recommend rephrasing these lines by providing a balancing statement for both statistical and advanced modeling techniques.

Line 239-243: Here authors are discussing the possible ways of improving the FAO's irrigated area datasets. Combination of statistical data and remote sensing could be an effective approach for upgrading to reporting mechanisms of irrigated areas at country level.

Conclusion

Please add conclusion in the manuscript along with the future directions.

Reviewer #2 (Remarks to the Author):

Review of

"Irrigated areas drive irrigation water withdrawals"

This paper argues that irrigation water withdrawal at the country level can be estimated from spatial data on irrigated areas alone (such as the global map of irrigated areas (GMIA), and that more complex approaches that use other data and models can be greatly simplified. This is based on an analysis of modelled irrigation water withdrawal from 8 global hydrological models and a linear regression with reported data from FAO AQUASTAT, which has its own limitations, as the authors acknowledge.

The method is a simple linear regression analysis of irrigated areas and reported water withdrawal for 8 models. It is well known that large uncertainties exist in irrigated areas previous studies have investigated the uncertainties related to irrigated areas and their impact on withdrawal estimates (for example, (Wisser et al., 2008), <https://doi.org/10.1029/2008GL035296>).

It is a trivial result and not surprising that irrigated areas and water withdrawal are well correlated. It is also not surprising the global model have very similar estimates for irrigation water withdrawal when they use the same map of irrigated areas. In fact, many of the models have very similar sub-modules to compute irrigation water demand and withdrawal, use the same parameterization of crops, and the same assumptions on irrigation efficiency. For some of the models that are analyzed here, the results are probably taken from model runs that were driven by the same climate input data, as part of the ISI-MIP project.

Also, most models were "adjusted" to match the reported numbers in AQUASTAT, even if they are not calibrated in a strict sense to match those values.

I agree that using the GMIA would remove some of the complexity of the GM but I am not sure that the irrigation modules are really relevant in term of computing, compared to cloud or evaporation, or percolation processes that are considered in many of those models. Furthermore, other data set that

are included in the models now (crop rotation, planting data, and others) are continuously being improved and this improvement will eventually improve estimates of irrigation water withdrawal.

In conclusion, I think the study adds very little to improving estimates on irrigation water withdrawal and I don't see enough novelty or extreme importance that would be required for a publication in Nature communications.

Reviewer #3 (Remarks to the Author):

Review of the manuscript "Irrigated areas drive irrigation water withdrawals" for Nature Communications

Using a wide range of output data from global models and FAO-based databases, the authors show that irrigation water withdrawals are largely explained by irrigated areas only and that the function governing its relationship is linear. The authors reflect on the advantages of emulators to describe in a simple and meaningful way complex non-linear systems, and advocate for a better recording and representation of the extent of irrigated areas.

I found the paper quite interesting and revealing. The analysis finds a strong relationship between irrigation water withdrawals and irrigated areas at different geographical scales. However, there are several aspects about this relationship that need more consideration and further discussion. First, the static-dynamic implications of this relationship. The analysis is based on 2005/2010 data and describes quite good that period, however the paper provides little insights on how irrigation water withdrawals will change in the future due to climate change. We know that climate change will hit regions unevenly and some dry and wet scenarios are expected. How good is this relationship expected to capture future irrigation water withdrawals in a world with climate change? Second, giving the importance of irrigated areas on determining future irrigation water withdrawals, a discussion on the drivers behind the change in irrigated areas is needed. For example, how are climate change impacts going to be factored on the dynamics of irrigated areas? This could easily be the topic of another paper, but is the simple method described in the paper robust to predict future water withdrawals in a changing climate? Or do we still need complex models to assess how future changes in temperature and effective precipitation are going to impact irrigated areas? Indeed, climate change that imposes serious threats to global food production and food security is only mentioned once in the text. Third, as the relationship is obtained at the continental level, meaning that the analysis is based on averages and it is highly aggregated, it fails in capturing the heterogeneity of irrigation systems among countries and within countries. A critical discussion around this is needed. For example, why the relationship was not assessed at the river basin or agro-ecological zone level, where similar biophysical conditions are expected? At that level, I would expect a strong influence of biophysical factors on determining irrigation water withdrawals.

In general, one would expect Europe as the continent with most reliable data. However, for most of the GM models Europe shows the weakest relationship among all continents (Fig 2). Why is this happening? Would this compromise the robustness of the method?

Specific comments:

- The motivation of the study is not well supported. The authors could stress on the importance and role of irrigation for food security in a changing climate.
- While in general the manuscript is well written, there are some typos in the document. A careful proof-reading is necessary. For example, see paragraph starting in line 58.
- The text and Fig 1 should clearly indicate that irrigation water withdrawals and irrigated areas have different data years.

- Please provide a description of the acronyms used such as GHM, LSM, LESM
- The sentence in 263-365 is not clear
- Line 306: It will be very useful for general readers to describe what imputation stand for in statistics in simple terms, something like "imputation is the process of replacing missing data with substituted values". The text starting in line 310 needs to be better connected to the 4 uncertainty tested. This could be done, in most of the cases, just adding the trigger that is referred to in parenthesis.
- In fact, the uncertainties in the materials and methods are first presented as numbers (1,2,...) then as triggers (X1, X2,...). They need to be consistent along the text and use them to clarify the explanation in the text. For example, when the authors talk about "the sample selected from this distribution ..." and "the iteration retrieved", it appears to be 2 different things. If both are referred to as X3, then it is clear for the reader.
- The four uncertainties need to be clear from the beginning of the document. The information presented in lines 334-338 helps to characterize the uncertainties, but it is quite late in the text.
- From 338 to 345 the description is too technical and specialized to follow. It would be appreciated if the authors add more details and relate the concepts to the specific problem setup.
- What is v , N_t ? Why 13?
- The decomposition analysis needs to be better integrated to the analysis. The meaning of the variables needs to be discussed.

Reply to the Reviewers of
Irrigated areas drive irrigation water withdrawals
(NCOMMS-20-48716-T)

Arnald Puy, Emanuele Borgonovo, Samuele Lo Piano, Simon A. Levin, Andrea Saltelli

May 2, 2021

Authors' comments are in blue.

Reviewer #1

Comment 1. The manuscript "Irrigated areas drive irrigation water withdrawals" aims at assessing the efficacy of irrigated areas data to estimate the agricultural water use at continental scale using statistical approach. This work is novel and it has great potential to influence the conventional thinking in global information systems on Ag water. The methodology employed by the authors using Aquastat, FAO-GMIA and multiple hydrological models to calculate the gridded water withdrawals, in general, seems sound and interesting. I found this work impressive, and would definitely like to admire the efforts the authors put together for collecting, arranging, and analyzing the data. Though the sophisticated models for precise estimation of water use at large scale are inevitable, statistical approaches could provide a great advantages in terms of time and computational expense. Also, it is more convenient to communicate with the policy makers when moderately complex estimation techniques are used that are linked with widely used datasets e.g., FAO-GMIA. The manuscripts is well-written and flow of information is smooth; however, some improvements are needed before it can be recommended for publication. I would suggest authors to submit the revised version of the manuscript so it can be further reviewed. Following are my comments and suggestions for possible improvements:

Thank you very much for your detailed review, which has helped us improve the manuscript significantly. Below we provide a detailed account on how we have addressed your observations in the new version of the piece.

Introduction

Comment 2. Please elaborate the purpose of this study in a separate paragraph. You have mentioned the two approaches for calculating global irrigation water withdrawals and described some of their shortcomings. It will be interesting to see what kind of gaps other researchers have highlighted related to irrigation water

withdrawals estimations at global scale. You can link you literature review with the challenges in policy making based on robust knowledge, and substantiate the need of establishing less-complex and time-saving methods to evaluate the water use for irrigating crops.

Thanks for this comment. We have added the following paragraph in the upgraded version of the manuscript, after discussing the drawbacks of the FAO's and the GM's approach (L. 60–69):

These drawbacks are amplified by uncertainties in crop types, growing seasons, agrarian practices, irrigated areas and local soil and climatic conditions [1]. Global irrigation water withdrawal estimates are therefore highly sensitive to the selection of the FAO's or the GM's approach, and even the choice of a specific GM is a source of bias [2]. The reliance on multi-model ensembles of GMs allows to obtain probabilistic estimates yet exacerbates the computational, opacity and uncertainty-related problems mentioned above [3]. Such flaws limit the utility of global irrigation water withdrawal estimates in the policy realm, where stakeholders and non-experts alike should be able to swiftly replicate the results or, at least, understand the main assumptions upon which the analysis is based [4–6].

Results

Comment 3. The results section is well elaborated and provides a good insight into the correlation of irrigated areas and water withdrawals. Authors included all major countries that practice irrigation worldwide but I was not able to find China in the text, though it is shown in figure 4a. Authors are encouraged to expand their findings from China and India as well since they are world's two largest countries with irrigation infrastructure. As authors might have seen in the literature that in China and India there are higher uncertainties in irrigated area reporting, and it would be suitable to provide more details in the results.

The manuscript now explicitly mentions China and India in the sub-sections within the Results (L. 142–144; 180). We have also added a paragraph in the Discussion section where we discuss the uncertainty in the reported irrigated areas for these countries, as well as its implications for the global modeling of irrigation water withdrawals. It reads as follows (L. 241–251):

Let us illustrate this issue with the case of China and India, for instance. The FAO-GMIA reports for both countries irrigated areas extending over 61 Mha. These point estimates turn into ranges that respectively span 43–74 Mha and 15–88 Mha if the other irrigated area datasets are taken into account [7–9]. These divergences are explained by the different methodological approaches mobilized to map irrigated areas, including the definition of what is considered an “irrigated area” and the degree of reliance on official statistics. Given the strong weight of the extension of irrigation, the already large variance in irrigation water withdrawals displayed in Figs. 2–3 will get much larger if GMs factor this uncertainty in. The same will apply to the estimates of future irrigation water withdrawals, for the uncertainty of global irrigated areas in 2050 spans half an order of magnitude (300–800 Mha, with the most extreme values reaching 1,800 Mha) [10].

Comment 4. Line 143-154: Authors are explaining the interlinkage of irrigation water withdrawal values and the fraction of irrigated area in a cell. This is an important component of the results section and needs more attention. Therefore, I would suggest that authors add details on how the fraction is estimated in models

and what systematic errors or discrepancies are associated with it. Also, if the models incorporate FAO-GMIA’s gridded fraction for irrigated land, it should be clearly mentioned.

Thanks for this comment. To avoid confusion, in the new version of the manuscript we no longer use the fraction of each cell that is irrigated, but the total extension of irrigation in each cell (in hectares). The conversion has been done by taking each pixel given the coordinates and spatial resolution of the map, re-projecting to get the total area of the pixel (in ha), and then calculating the total extension of irrigation from the fraction of the pixel that is irrigated. We hope that this improves the consistency and intelligibility of the manuscript, for now we only discuss irrigated areas (and not the fraction of the cell that is irrigated).

Following Reviewer #1’s comment, we stress that GMs incorporate the gridded version of the FAO-GMIA map at the beginning of the results section (L. 81–83):

Irrigated areas in GMs are parametrized with the Global Map of Irrigated Areas (FAO-GMIA) [11], a gridded product that documents the extension of irrigation at a 5 arcmin resolution.

Comment 5. Line 155-172: Authors have mentioned that the slope of the linear regression of $\log(y)$ and $\log(x)$, denoted as β , which may be governed by the size of the irrigation schemes and complexity for estimating water withdrawals. I suggest authors to explore more factors that might affect this scaling framework. Only linking the β with the size of the irrigation scheme and calculation complexity may not be adequate from scientific point of view. Water productivity of the irrigated agriculture could be another factor. Furthermore, water laws of a country or state could play a role. Also, authors need to consider the fact that large irrigation areas may not have disproportionately higher evaporation, for example, large center pivot irrigated areas.

Thanks for this observation. In our paper we show that almost all the variation in irrigation water withdrawals is explained by a single parameter, the extension of irrigation. In that sense, the scaling relationship between both variables is tight, and is summarized by the scaling exponent β .

In the previous version of the manuscript we also examined the influence of factors such as potential evaporation, total evapotranspiration or irrigation efficiencies, and found that they are of either no significance or of very minor relevance compared to irrigated areas.

Following Reviewer’s #1 observation that water productivity could also be a relevant factor, we retrieved worldwide water productivity data from the World Bank at <https://data.worldbank.org/indicator/ER.GDP.FWTL.M3.KD>, and plotted it against irrigation water withdrawals. As displayed in Figure R1 below, the lack of a trend rules water productivity out as a factor significantly co-evolving with irrigation water withdrawals.

In order to clarify scaling, we have added the following paragraph in the Discussion section of the new manuscript (L. 210–218):

That a complex variable such as the volume of water withdrawn for irrigation is nicely described by just a single factor may appear surprising given the large space of relevant factors influencing its behaviour (e.g., crop type and calendars, growing seasons, irrigation efficiency, climate, crop evapotranspiration, soil texture, etc). Yet several degrees of freedom are often determined or summarised by a small set of constraints or even by a single parameter. Size is one of such parameters: for animals, it defines their strength, metabolic rate, life span or population density [12, 13]; for cities, its pace of innovation, number of patents or total electrical consumption [14, 15]. The size of irrigated areas (e.g. their extension) appears to be a similar driving force for

irrigation water withdrawals..

We believe Reviewer #1 is right when they propose water laws as a potentially relevant factor. However, we are afraid that a thorough investigation of this relationship is beyond the scope of our manuscript, which is predominantly quantitative in nature and framed as a communication. We thank Reviewer #1 for raising this issue and leave the investigation of this link for future manuscripts.

With regards to large irrigation areas not having a disproportionately higher evaporation: we agree with Reviewer #1 that this might not always be the case. To avoid confusion, we have cleared the sentence where we suggest this connection from the new version of the manuscript.

Figure R1: Scatterplot of irrigation water withdrawals against water productivity. Each dot is a country.

Discussion

Comment 6. Line 205-212: Do authors mean statistical emulators here? If that’s the case, then it should be clearly stated. Authors should realize that it could raise further questions on the reliability of the mechanistic models if sub-models are replaced by the simple algorithms. Despite the fact that complexity leads to extensive computation, it still allows the modelers to keep a track of variables that have higher influence on the outputs. Though this study argues about the efficacy of the statistical methods that are time-saving, the advancement in modeling and benefits of assimilation of complex models with satellite based datasets cannot be ignored. Therefore, adding conceptual depths in models cannot be labeled as dubious pursuit just by skewing the discussion towards uncertainty in irrigated area data. I would recommend rephrasing these lines by providing a balancing statement for both statistical and advanced modeling techniques.

Thanks for this comment. The paragraph has been nuanced in the upgraded version of the manuscript, and now reads as follows (L.219–230):

The irrigation module of GMs contains parameters or secondary outputs whose influence in the calculation of water withdrawals is very minor or inconsequential [i.e. total evapotranspiration, potential evaporation or irrigation efficiency once controlled for irrigated area, Figs. S12–S14]. The effect of parameters such as crop

coefficients or crop calendars, which vary as a function of time, or of complex sub-models such as fertilization schemes or the climate driver [16–19], is much harder to check. However, if a linear regression can nicely fit the input-output mapping of complex hydrological algorithms, other similar fast-running statistical emulators may be able to work in the computationally expensive sub-models nested within GMs. Substituting these sub-models with time-effective emulators can be an effective way to save computational resources and bridge model realism with computational efficiency. This may allow modelers to focus on answering relevant water policy questions without the extra burden of managing unneeded complexity.

Conclusion

Comment 7. Please add conclusion in the manuscript along with the future directions.

The structure of an Article, according to *Nature Communications*, does not include a conclusion section. We paste the relevant paragraph where this is stated, extracted from <https://www.nature.com/ncomms/submit/article>:

The main text of an Article should begin with a section headed Introduction of referenced text that expands on the background of the work (some overlap with the abstract is acceptable), followed by sections headed Results, Discussion (if appropriate) and Methods (if appropriate). The Results and Methods sections should be divided by topical subheadings; the Discussion should be succinct and may not contain subheadings.

This notwithstanding, and in order to address this recommendation by Reviewer #1, we have added a closing paragraph in the Discussion section in which we also outline possible future research threads. It reads as follows (L. 284–296):

(...). An extension of our work is to explore why the trend between irrigated areas and water withdrawals is the weakest in European countries. Another direction is to assess whether irrigated areas drive irrigation water withdrawals in other irrigation systems and/or in different sub-regional contexts. This will help assess the extent to which this trend scales down in a robust way.

Finally, we should stress that we are not proposing to substitute approaches relying on crop, soil and climate parameters with linear regressions of irrigated areas. At small granularities, e.g., the plot or the scheme level, these methods nicely appraise physical process and facilitate monitoring of water withdrawals through time. What we argue is that so much detail may not be warranted at larger scales or under strong uncertainties. The debate between proponents of computationally-intensive methods and advocates of more simple approaches is vibrant in the climate modeling community [20, 21]. Both parties also need to be heard in the field of global hydrology.

Reviewer #2

Comment 8. This paper argues that irrigation water withdrawal at the country level can be estimated from spatial data on irrigated areas alone (such as the global map of irrigated areas (GMIA), and that more complex approaches that use other data and models can be greatly simplified. This is based on and analysis of modelled

irrigation water withdrawal from 8 global hydrological models and a linear regression with reported data from FAO-AQUASTAT, which has its own limitations, as the authors acknowledge.

Thanks for reviewing our manuscript. Below we provide a point-by-point reply to the comments by Reviewer #2.

Comment 9. The method is a simple linear regression analysis of irrigated areas and reported water withdrawal for 8 models. It is well known that large uncertainties exist in irrigated areas; previous studies have investigated the uncertainties related to irrigated areas and their impact on withdrawal estimates (for example, Wisser et al., 2008, <https://doi.org/10.1029/2008GL035296>).

We agree with Reviewer #2 in that linear regressions are simple models. In our case, the use of a linear regression to model the relation between irrigated areas and irrigation water withdrawals is determined by the trend displayed by the data (see Figures R2–R7 below and answer to comment 15). Given this pattern, the choice of a linear model appears highly reasonable.

Our study focuses on eight GMs and two FAO-based datasets, and combines linear regressions with thorough, state-of-the-art uncertainty and sensitivity analysis. This technical aspect, although key, appears to be overlooked by Reviewer #2 in their review.

With regards to the Wisser et al. [22] paper: indeed, Wisser et al. observed that the water withdrawal estimates yielded by WBM_{plus} significantly change when the IWMI-GIAM (rather than the FAO-GMIA) is used to parametrize irrigated areas. Such sensitivity would imply that GMS ought to incorporate irrigation modules that can easily switch from one irrigation map to the other, so the estimates produced are robust against the uncertainty in the extension of irrigation. To our knowledge, this key insight of Wisser et al. was never translated into practice and most GMs continued relying exclusively on the FAO-GMIA, until today.

A full investigation of the reasons behind such lock-in despite this evidence of bias is beyond the scope of our manuscript. Here we just venture one possible explanation: once established, complex models such as GMs lack the flexibility to run under different assumptions simultaneously.

On that note, our study makes three new contributions:

1. It shows empirically and at different scales the extent to which irrigated areas drive irrigation water withdrawals.
2. It proves that the irrigation module of GMs errs on the side of complexity, for the simulated irrigation water withdrawals are largely driven by the extension of irrigation, parametrized by the FAO-GMIA.
3. It offers an empirically-grounded, agile and manageable method to compute global irrigation water withdrawals –which, unlike that used by GMs, allows to easily integrate any of the five irrigated area datasets currently available, thus permitting a swift and thorough exploration of the model’s uncertain space.

Comment 10. It is a trivial result and not surprising that irrigated areas and water withdrawal are well correlated.

Indeed, the relation may appear trivial but it is of consequence, yet it has been overlooked by GMs all the same (they are grounded on several physical equations and dozens of parameters), as well as in the interpretation

Figure R2: Scatterplots of irrigated areas against irrigation water withdrawals at c. 2005. Irrigated area data are provided by the FAO-GMIA [23]. Irrigation water withdrawals are retrieved from GMs (CLM45, DBHM, H08, LPJmL, MPI-HM, PCR-GLOBWB, VIC, WaterGap) and from FAO-based datasets (Aquastat, Liu et al. [24]). Each dot is a country.

of their results. As we showed in the previous version of the manuscript (Figures S12–S43), and despite their computational complexity, several GMs behave at the grid cell level like linear models driven by irrigated areas. We attach below one of the most significant plots supporting our case (Figure R3); the rest can be found in the Supplementary Materials.

Comment 11. It is also not surprising the global model have very similar estimates for irrigation water withdrawal when they use the same map of irrigated areas. In fact, many of the models have very similar sub-modules to compute irrigation water demand and withdrawal, use the same parameterization of crops, and the same assumptions on irrigation efficiency. For some of the models that are analyzed here, the results are probably taken from model runs that were driven by the same climate input data, as part of the ISI-MIP project.

Also, most models were “adjusted” to match the reported numbers in AQUASTAT, even if they are not calibrated in a strict sense to match those values.

We amicably disagree with Reviewer #2. GMs do not have very similar estimates of irrigation water withdrawals: a swift look at Figures 3–4 of the original manuscript evidences that, depending on the GM used, irrigation water withdrawals at the country level can vary by several orders of magnitude. This means that the selection of a specific GM for the production of irrigation water withdrawal values is already a source of bias that can misguide policies on water security. Our approach overcomes this limitation: by combining linear regressions with uncertainty / sensitivity analysis, we produce a range of values that sharply frame the estimates of eight GMs plus two FAO-based datasets combined, yet in a much more efficient way. In the upgraded version of the manuscript we include these ranges as a .csv file.

With regards to Reviewer’s #2 comment on the use of model runs with the same climate input data: indeed, in the original manuscript we focused on simulations relying on the same climatic settings to facilitate the comparison between model runs. We used WFDEI climate data except for MPI-HM and CLM45, which relied on MIROC5 and GFDL respectively. This was stated in lines 266–268 of the original manuscript.

In order to expand the range of studied climatic scenarios, in the upgraded version of the manuscript we also

Figure R3: Relation between irrigation water withdrawals and irrigated areas at the grid cell level for Africa. The GM model used is CLM45.

examine the trend between irrigation water withdrawal and irrigated areas under future climate change (see Figures R8–R9 below). In this specific case, the model runs are driven by MIROC5 under different climatic and social assumptions. See our reply to Comment 15 for more details. Our results also hold under these different climatic scenarios.

As for the adjustment to the numbers reported by AQUASTAT: this was indeed the case for WaterGap, LPJmL, PCR-GLOBWB and H08. For these models, we used the datasets generated by Huang et al. [25], who downscaled the data yielded by these four GMs with AQUASTAT data. This was stated in lines 260–263 of the original manuscript. Such downscaling, which merged two different sources of water withdrawal data (official FAO-based statistics and GM-produced data), aimed at providing a more comprehensive account of water use by agriculture, and is common practice in global hydrological modeling.

Comment 12. I agree that using the GMIA would remove some of the complexity of the GMs but I am not sure that the irrigation modules are really relevant in term of computing, compared to cloud or evaporation, or percolation processes that are considered in many of those models. Furthermore, other data set that are included in the models now (crop rotation, planting data, and others) are continuously being improved and this improvement will eventually improve estimates of irrigation water withdrawal.

Thanks for this comment. We believe Reviewer #2 argues that limiting the analysis to the irrigated areas of the FAO-GMIA will just moderately simplify GMs, since other modules (e.g. cloud, evaporation, percolation processes) are also complex and highly relevant.

In the previous version of the manuscript we showed that these other modules (e.g. evapotranspiration, potential evaporation, irrigation efficiencies) are not really needed to simulate irrigation, for their effect on irrigation water withdrawals is either negligible or very minor compared to irrigated areas (see Figures S12–S14 of the Supplementary Materials).

As for the assumption “that ongoing research will improve existing data sets, ultimately leading to more accurate irrigation water withdrawal estimates”, we would like to make the following comments:

Firstly, depending on how this “improvement” is done, uncertainty may actually increase: the addition of new terms and processes to improve the model’s descriptive capacity often leads to larger errors once uncertainties are thoroughly examined, a phenomenon known as the “O’Neill conjecture” [26, 27]. Secondly, more research can lead to larger, and not smaller, uncertainty ranges: an example are IPCC’s models on climate change, whose uncertainty ranges have been increasing over the years despite the massive allocation of funds to better understand complex climate processes [28]. Thirdly, remote sensing techniques may improve, yet they are unlikely to become “error-free”. Crop rotation, or planting data maps, as all remote sensing-based products, incorporate topographic effects, sensor noise, conversion errors and several other misrepresentations derived from our incapacity to perfectly model reality, including scientists’ values and viewpoints [29]. Assuming that this might disappear with further research is questionable. In any case, our analysis does not discourage scientists from pushing towards this direction.

Finally, it is unclear how much research is needed to achieve more accurate estimates of irrigation water withdrawals. The marginal benefits of improving models through the addition of sub-routines or the inclusion of new mapping technologies can be zero or even negative [30], draining personal and financial resources with scarce results. By evidencing that almost all is driven by irrigated areas, our research efficiently channels future

research and provides a robust stopping-point to the quest for “better” estimates: it is doubtful to focus on improving crop pattern representation, planting data or any other process without having a more appropriate mapping of irrigated areas.

Comment 13. In conclusion, I think the study adds very little to improving estimates on irrigation water withdrawal and I don’t see enough novelty or extreme importance that would be required for a publication in *Nature Communications*.

We hope that the extent of our contribution is now clearer to Reviewer #2.

Reviewer #3

Comment 14. I found the paper quite interesting and revealing. The analysis finds a strong relationship between irrigation water withdrawals and irrigated areas at different geographical scales. However, there are several aspects about this relationship that need more consideration and further discussion.

Thank you very much for your nice words on our paper and for your suggestions, which have significantly improved the manuscript. Please see below a detailed account on how we have addressed your comments in the upgraded version of the paper.

Comment 15. First, the static-dynamic implications of this relationship. The analysis is based on 2005/2010 data and describes quite good that period, however the paper provides little insights on how irrigation water withdrawals will change in the future due to climate change. We know that climate change will hit regions unevenly and some dry and wet scenarios are expected. How good is this relationship expected to capture future irrigation water withdrawals in a world with climate change?

Thank you very much for these comments. Indeed, the original manuscript provided a screenshot of the relation between irrigated areas and irrigation water withdrawals in 2005/2010. We selected this period for two reasons: 1) it is the closest period for which there is solid data on both irrigated areas and irrigation water withdrawals, and 2) it perfectly summarizes the historical relation between both variables, since the strong relationship attested in 2005 goes back to at least 1900 (Figures R4–R7).

To make this clearer, in the new manuscript we have included a few sentences informing of the historical robustness of this relation and some lines explaining why we focus on the 2005/2010 period only (L. 83–90). We have also added Figures R4–R7 in the upgraded Supplementary Material.

In order to check whether irrigated areas can also be used to predict future irrigation water withdrawals under climate change, we have extended our analysis as follows:

1. We have retrieved from the ISIMIP webpage the projections of irrigation water withdrawals to 2050 made by four GMs (PCR-GLOBWB, LPJmL, H08 and MPI-HM) under the following social, climatic and CO₂ scenarios:
 - *rcp26/rcp26*: Water abstraction and land use (including irrigated areas) change according to the Shared Socioeconomic Pathway 2 (SSP2, “Middle of the road”). Future climate and CO₂ concentration evolve as outlined by the Representative Concentration Pathway 2.6 (RCP2.6).

- *rcp60/rcp60*: Water abstraction and land use (including irrigated areas) change according to SSP2. Future climate and CO₂ concentration evolve as outlined by the Representative Concentration Pathway 6.0 (RCP6.0).
- *2005soc/rcp26*: Land use (including irrigated areas), nitrogen deposition and fertilizer input are fixed at 2005 values. Future climate and CO₂ concentration as in RCP2.6.
- *2005soc/rcp60*: Land use (including irrigated areas), nitrogen deposition and fertilizer input are fixed at 2005 values. Future climate and CO₂ concentration as in RCP6.0.
- *2005soc/rcp85*: Land use (including irrigated areas), nitrogen deposition and fertilizer input are fixed at 2005 values. Future climate and CO₂ concentration as in RCP8.5.

2. We have checked whether these future projections fall within the irrigation water withdrawal ranges produced with our regression-based approach. We have assumed that irrigated areas are able to capture future irrigation water withdrawals under climate change if most of the GMs' estimates are bounded by our estimates.

The results indicate that most of the estimates yielded by GMs in a context of climate change can simply be obtained as a function of the irrigated areas reported by the FAO-GMIA (Figures R8–R9). This finding significantly strengthens the main message of the paper: irrigation water withdrawals are largely driven by irrigated areas alone.

These results have been incorporated in the upgraded version of the manuscript (L. 152–161).

PCR-GLOBWB

Figure R4: Historical relationship between irrigated areas and irrigation water withdrawals (1900–2000). Data on historical irrigated areas were retrieved from the Historical Irrigation Data set (HID) [31], which follows the FAO-GMIA methodology. Data on historical irrigation water withdrawals was retrieved from ISIMIP [32].

MPI-HM

Figure R5: Historical relationship between irrigated areas and irrigation water withdrawals (1900-2000). Data on historical irrigated areas were retrieved from the Historical Irrigation Data set (HID) [31], which follows the FAO-GMIA methodology. Data on historical irrigation water withdrawals was retrieved from ISIMIP [32].

Comment 16. Second, giving the importance of irrigated areas on determining future irrigation water withdrawals, a discussion on the drivers behind the change in irrigated areas is needed. For example, how are climate change impacts going to be factored on the dynamics of irrigated areas? This could easily be the topic of another paper, but is the simple method described in the paper robust to predict future water withdrawals in a changing climate? Or do we still need complex models to assess how future changes in temperature and effective precipitation are going to impact irrigated areas? Indeed, climate change that imposes serious threats to global food production and food security is only mentioned once in the text.

Thank you very much for these comments. Indeed, the impact of climate change in the future evolution of irrigated areas is a topic that goes beyond the purpose of our manuscript. We know, however, that the extension of irrigation changes depending on factors such as water availability (directly related to climate), taxation or population growth; see, for instance, Puy et al. [10, 34] and Puy [35]. The behaviour of these parameters is very difficult to predict, which hampers obtaining accurate estimates of the future extension of irrigation. In fact, once basic uncertainties are factored in, the extension of global irrigated areas in 2050 can span half an order of magnitude (300–800 Mha) [10].

As for the question on whether our method can forecast water withdrawals under climate change: in Figures R8-R9 we show that a regression yields for 2050 almost the same irrigation water withdrawal estimates as those predicted by four GMs under four different climatic scenarios. This makes our method at least as robust as that of GMs yet much simpler, transparent, efficient and understandable.

Complex models are therefore not necessarily better than simple ones. In fact, excess complexity might be a hindrance: high-dimensional models with dozens of parameters and processes tend to be so computation-

Figure R6: Historical relationship between irrigated areas and irrigation water withdrawals (1900-2000). Data on historical irrigated areas were retrieved from the Historical Irrigation Data set (HID) [31], which follows the FAO-GMIA methodology. Data on historical irrigation water withdrawals was retrieved from ISIMIP [32].

ally expensive that a thorough exploration of their uncertainty space is often unaffordable. This can lead to artificially-precise model outputs that might misguide policy-making. Complex models are also rigid and lack the agility to work in a “what if” configuration. In our manuscript we are offering a much simpler alternative that produces the same results as eight GMs combined, as well as the possibility of comprehensively navigating the relative importance of different assumptions.

Following the comments of Reviewer #2, in the new version of the manuscript we have strengthened the links of our study with food security and climate change, especially in the Introduction section (see our reply to comment 19 below.)

Comment 17. Third, as the relationship is obtained at the continental level, meaning that the analysis is based on averages and it is highly aggregated, it fails in capturing the heterogeneity of irrigation systems among countries and within countries. A critical discussion around this is needed. For example, why the relationship was not assessed at the river basin or agro-ecological zone level, where similar biophysical conditions are expected? At that level, I would expect a strong influence of biophysical factors on determining irrigation water withdrawals.

Thank you. We agree with Reviewer #3 in that our analysis was mostly based on the aggregation of values at the country level. However, we also checked whether the trend between irrigation water withdrawals and irrigated areas emerged at smaller scales. To that aim, we used the data at the irrigation system level from the Australian National Committee on Irrigation and Drainage (ANCID) [36, 37], as well as the data collected at the grid cell level from the GMs.

Figure R7: Historical relationship between irrigated areas and irrigation water withdrawals (1900-2000). Data on historical irrigated areas were retrieved from the Historical Irrigation Data set (HID) [31], which follows the FAO-GMIA methodology. Data on historical irrigation water withdrawals was retrieved from ISIMIP [32].

In the new version of the manuscript we have expanded the scale of analysis following this comment by Reviewer #3, so we capture the relation between irrigated areas and irrigation water withdrawals in a broader range of settings, e.g. within countries and among countries.

Firstly, we have checked whether this relation holds within countries by using the data retrieved by Solley et al. [38] for each State of North America. Secondly, we have also assessed whether the trend applies at the State level with the data compiled by Ivahnenko and Flynn [39] for the Colorado county. The results are presented in Figure R10 and reinforce the existence of a strong link between both variables regardless of the scale of analysis. We have included the new plot and extended this discussion in the upgraded version of the manuscript (L. 162–181).

As for the collection of data at the River-basin or agro-ecological zone level: irrigation water withdrawal estimates at these scales are often produced by aggregating values obtained at the grid cell level (see for instance [40, 41]). If the trend between water withdrawals and irrigated areas holds at the grid cell level, it is very likely that it will emerge at the river basin level as well. In the previous version of the manuscript, we showed that water withdrawal estimates at the grid cell level are indeed largely driven by irrigated areas (see Figures S12–S43 of the Supplementary Materials).

Following Reviewer #3’s suggestion, we have clarified this issue in the upgraded version of the manuscript (L. 178–181).

Comment 18. In general, one would expect Europe as the continent with most reliable data. However, for most of the GMs models Europe shows the weakest relationship among all continents (Fig 2). Why is this

happening? Would this compromise the robustness of the method?

This is a very good remark. Indeed, Europe presents the weakest relationship between irrigated areas and irrigation water withdrawals.

We venture that this might be related to the substantial reliance of European countries on modern irrigation technologies. Under traditional flood irrigation there is a correspondence between the size of the irrigated area and the volume of water used for irrigation. This correspondence breaks down under drip or sprinkler irrigation, for these water-saving devices allow farmers to irrigate an area much larger than the area they would be able to irrigate under flood irrigation (assuming that the same volume of water used under drip is used under flood-irrigation). In general, irrigated areas under drip irrigation use less water than what they are supposed to given their size. Investigating whether this explains the weaker relation between irrigated areas and irrigation water withdrawals in Europe is a very interesting research pursuit, yet the topic falls beyond the scope and the size limitation of the present paper. In the upgraded version of the manuscript, however, we propose this research thread as a possible future line of enquiry.

In any case, this weaker trend attested for Europe does not compromise the robustness of our method: it only leads to slightly wider confidence intervals for European countries, as one can see in Figure 4 of the upgraded manuscript.

Specific comments

Comment 19. The motivation of the study is not well supported. The authors could stress on the importance and role of irrigation for food security in a changing climate.

Thank you for this suggestion. We have rewritten the Introduction following this advice, and the introductory paragraphs now read as follows (L. 27–43):

Irrigation agriculture is at the forefront of global food security. With the potential to provide crop yields more than two times as large as dryland agriculture [43, 44], irrigation agriculture currently produces approximately 40% of all food consumed worldwide in just 20% of the total cultivated land [45]. Its capacity to maximize yields per unit of terrain is conditional upon the investment of high labour inputs per surface unit and the provision of a steady freshwater supply, which relaxes the dependency of crops on rainwater seasonality [46, 47]. This allows year-round harvests while reducing adverse impacts on crops from dry spells. Such features make irrigation agriculture a key resource to buffer population growth in a context of climate change.

The demand of water by irrigation has been constantly rising over the last decades [48]. In general, it is expected to increase even further in the next years due to changes in precipitation patterns, higher temperatures and the expansion of irrigated areas to meet the projected boost in food demands [2, 10, 49–52]. Getting reliable estimates of irrigation water withdrawals is thus regarded as a first step towards a more informed management of global freshwater resources [53, 54], ultimately endowing us with better tools to ensure food security without damaging the water system. At present, there are two main approaches to calculate global irrigation water withdrawals (...)

Comment 20. While in general the manuscript is well written, there are some typos in the document. A careful proof-reading is necessary. For example, see paragraph starting in line 58.

Thank you. The upgraded version of the manuscript has undergone a professional proof-read. We hope that typos, misspellings and grammatical error are no longer present in the resubmitted version.

Comment 21. The text and Fig 1 should clearly indicate that irrigation water withdrawals and irrigated areas have different data years.

Thank you for this remark. In the upgraded version of the manuscript we have fixed this issue: all the irrigation water withdrawal data retrieved from GMs come from 2005 to match the FAO-GMIA, which reflects the extension of irrigation in 2005. As for the FAO-based datasets, they inform on irrigation water-withdrawals in 2012.

To clarify this issue, we have rewritten the caption of Figure 1 as follows:

Scatterplots of irrigated areas against irrigation water withdrawals. Irrigated area data are provided by the FAO-GMIA [23], which reflects the extension of irrigation in 2005. Irrigation water withdrawals are retrieved from GMs (CLM45, DBHM, H08, LPJmL, MPI-HM, PCR-GLOBWB, VIC, WaterGap) and from FAO-based datasets (AquaStat, Liu et al. [24]). GMs data is from 2005 and FAO-based data from 2012. Each dot is a country. The abscissa x and the ordinate y refer to Equation 1 (see Methods).

Comment 22. Please provide a description of the acronyms used such as GM, LSM, LESM

Thanks for this. In the new version of the manuscript we only refer to GMs. We describe the acronym in L. 52 of the updated version of the paper.

Comment 23. The sentence in 263-365 is not clear.

We assume that Reviewer #3 refers to the sentence in lines 263–265, as lines 363–365 include the “Code availability” section. The sentence in lines 263–265 of the previous manuscript now reads as

For DBHM, MPI-HM, VIC and CLM45 we use the data produced by the Inter-Sectoral Impact Model Inter-comparison Project (ISI-MIP) [32].

We hope its meaning is now clearer.

Comment 24. Line 306: It will be very useful for general readers to describe what imputation stand for in statistics in simple terms, something like “imputation is the process of replacing missing data with substituted values”. The text starting in line 310 needs to be better connected to the 4 uncertainty tested. This could be done, in most of the cases, just adding the trigger that is referred to in parenthesis.

We agree with Reviewer #3 in that the description of the imputation and the uncertainty and sensitivity analysis should be understandable to the general reader. In the new version of the manuscript (L. 355–439) we have rewritten the section and added more detail, better connecting the technical explanation with the study and with a clearer organization of the text. Now the “Uncertainty analysis” section starts with a pointy-by-point description of the triggers, which are always referred to by their abbreviation (X_1, \dots, X_4). We hope it is now clearer.

Comment 25. In fact, the uncertainties in the materials and methods are first presented as numbers (1,2,...) then as triggers (X_1, X_2, \dots). They need to be consistent along the text and use them to clarify the explanation in the text. For example, when the authors talk about "the sample selected from this distribution ..." and "the iteration retrieved", it appears to be 2 different things. If both are referred to as X_3 , then it is clear for the reader.

This has been corrected in the upgraded version of the manuscript. The triggers are now always referred to as X_1, X_2, \dots .

Comment 26. The four uncertainties need to be clear from the beginning of the document. The information presented in lines 334-338 helps to characterize the uncertainties, but it is quite late in the text.

Thank you. Now the characterization of the triggers is placed right at the beginning of the "Uncertainty analysis" section (L. 355).

Comment 27. From 338 to 345 the description is too technical and specialized to follow. It would be appreciated if the authors add more details and relate the concepts to the specific problem setup.

We agree that the description of the uncertainty and sensitivity analysis is a bit technical. In order to make it more understandable to the general reader, we have rewritten it, added more detail where needed, and tried to connect it better with our study. Please see lines 396-439 of the new manuscript. We have also included in the Supplementary Material a new figure to illustrate how the \mathbf{A} , the \mathbf{B} and the $\mathbf{A}_B^{(i)}$ matrices are constructed from the \mathbf{Q} matrix (see Figure R11 below).

Comment 28. What is v , N_t ? Why 13?

In the original manuscript, v indexed the row of the sample matrix used to conduct the uncertainty analysis, and N_t referred to the total number of model runs. This has been clarified in the upgraded version of the manuscript (L. 405-407).

As for the use of 2^{13} : this is the number of rows selected for the base sample matrix, which was constructed using Sobol' Quasi-Random Numbers [56]. The Sobol' sequence is a low-discrepancy sequence in base 2, and the number of rows in such a sequence needs to be a power of two to thoroughly explore the hyperspace. After some preliminary experiments, we observed that the use of 13 in the exponent ensured convergence of the Sobol' indices (See Figure 2c in the upgraded manuscript; the larger the sample size, the narrower the confidence intervals of the indices). This has been stated explicitly in the upgraded version of the manuscript (L. 398-401).

Comment 29. The decomposition analysis needs to be better integrated to the analysis. The meaning of the variables needs to be discussed.

Thank you for this observation. The notation for the variance-decomposition framework is now fully explained in the upgraded version of the manuscript (L. 420-439).

Irrigation water withdrawal ($10^9\text{m}^3/\text{year}$)

Figure R8: Comparison between our approach and the irrigation water withdrawals simulated by GM for 2050. The error bars and the black dots show the range and the median irrigation water withdrawal values obtained with our approach. The colored, shaped dots show the estimates produced by GM under five different social and climatic scenarios. Simulations with the “2005” suffix run on land use patterns (including the extension of irrigation) fixed at their 2005 values. Simulations that do not have the “2005” suffix assume that irrigated areas change according to the Shared Socioeconomic Pathway 2 (SSP2, “Middle of the Road”). rcp26, rcp60 and rcp85 refer to the Representative Concentration Pathway 2.6, 6 and 8.5 respectively [33].

Figure R9: Comparison between our approach and the irrigation water withdrawals simulated by GM for 2050. The error bars and the black dots show the range and the median irrigation water withdrawal values obtained with our approach. The colored, shaped dots show the estimates produced by GM under five different social and climatic scenarios. Simulations with the “2005” suffix run on land use patterns (including the extension of irrigation) fixed at their 2005 values. Simulations that do not have the “2005” suffix assume that irrigated areas change according to the Shared Socioeconomic Pathway 2 (SSP2, “Middle of the Road”). rcp26, rcp60 and rcp85 refer to the Representative Concentration Pathway 2.6, 6 and 8.5 respectively [33].

Figure R10: The trend between irrigated areas and irrigation water withdrawals scales down. a) Relation between both variables at different geographical scales. The leftmost plots show data for *circa* 60 Australian irrigation systems benchmarked between 1997–1998 and 2002–2003 [36, 37]. Each dot is an irrigation system. The rightmost plots present data on the extension of irrigation and irrigation water withdrawals across all counties in Colorado [39] and across all North-American States, retrieved respectively from Solley et al. [38] and Ivahnenko and Flynn [39]. Each dot is a County / State. b) Distribution of r^2 after bootstrapping ($R = 5,000$) the pooled data. The blue lines show the 95% confidence intervals, calculated with the bias-corrected and accelerated (bca) method [42]. See Fig. S21–S22 for the regression diagnostics.

$$\begin{aligned}
\mathbf{Q} &= \begin{pmatrix}
\overbrace{X_1 \ X_2 \ X_3 \ X_4}^{\mathbf{A}} & \overbrace{X_1 \ X_2 \ X_3 \ X_4}^{\mathbf{B}} \\
6 & 2 & 30 & 2 & 4 & 2 & 30 & 2 \\
3 & 1 & 3 & 1 & 8 & 1 & 18 & 1 \\
1 & 3 & 31 & 2 & 5 & 2 & 8 & 1 \\
8 & 2 & 17 & 1 & 6 & 2 & 40 & 2 \\
2 & 1 & 33 & 1 & 2 & 1 & 40 & 1
\end{pmatrix} \\
\mathbf{A}_B^{(1)} &= \begin{pmatrix}
4 & 2 & 30 & 2 \\
8 & 1 & 3 & 1 \\
5 & 3 & 31 & 2 \\
6 & 2 & 17 & 1 \\
2 & 1 & 33 & 1
\end{pmatrix} \\
\mathbf{A}_B^{(2)} &= \begin{pmatrix}
6 & 2 & 30 & 2 \\
3 & 1 & 3 & 1 \\
1 & 2 & 31 & 2 \\
8 & 2 & 17 & 1 \\
2 & 1 & 33 & 1
\end{pmatrix} \\
&\vdots
\end{aligned} \tag{1}$$

Figure R11: Example of the creation of an \mathbf{A} , \mathbf{B} and $\mathbf{A}_B^{(i)}$ matrices with $k = 4$. The \mathbf{Q} matrix has been created with Sobol' [55, 56] Quasi-Random numbers scrambled following Owen. We only show the first 5 rows of the Sobol' sequence. Note that the sequence has already been transformed in every column to the appropriate probability distribution of each trigger. The figure is based on Puy et al. [57].

References

- [1] United Nations. *The United Nations world water development report 2018: Nature-Based Solutions for Water*. 2018, pp. 1–139.
- [2] Y. Wada, D. Wisser, S. Eisner, M. Flörke, D. Gerten, I. Haddeland, N. Hanasaki, Y. Masaki, F. T. Portmann, T. Stacke, Z. Tessler, and J. Schewe. “Multimodel projections and uncertainties of irrigation water demand under climate change”. *Geophysical Research Letters* 40.17 (2013), pp. 4626–4632. DOI: [10.1002/grl.50686](https://doi.org/10.1002/grl.50686).
- [3] W. S. Parker. “Ensemble modeling, uncertainty and robust predictions”. *Wiley Interdisciplinary Reviews: Climate Change* 4.3 (2013), pp. 213–223. DOI: [10.1002/wcc.220](https://doi.org/10.1002/wcc.220).
- [4] A. Saltelli and S. Funtowicz. “When all models are wrong”. *Issues in Science and Technology* (2014), pp. 79–85.
- [5] A. Saltelli, A. Guimaraes Pereira, J. P. van der Sluijs, and S. O. Funtowicz. “What do I make of your Latinorum? Sensitivity auditing of mathematical modelling”. *International Journal of Innovation Policy* 9.2/3/4 (2013), pp. 213–234. DOI: [10.1504/IJFIP.2013.058610](https://doi.org/10.1504/IJFIP.2013.058610). arXiv: [1211.2668](https://arxiv.org/abs/1211.2668).
- [6] A. Saltelli, L. Benini, S. Funtowicz, M. Giampietro, M. Kaiser, E. Reinert, and J. P. van der Sluijs. “The technique is never neutral. How methodological choices condition the generation of narratives for sustainability”. *Environmental Science and Policy* 106. January (2020), pp. 87–98. DOI: [10.1016/j.envsci.2020.01.008](https://doi.org/10.1016/j.envsci.2020.01.008).
- [7] J. Meier, F. Zabel, and W. Mauser. “A global approach to estimate irrigated areas. A comparison between different data and statistics”. *Hydrology and Earth System Sciences* 22.2 (2018), pp. 1119–1133. DOI: [10.5194/hess-22-1119-2018](https://doi.org/10.5194/hess-22-1119-2018).
- [8] P. S. Thenkabail, C. M. Biradar, P. Noojipady, V. Dheeravath, Y. Li, M. Velpuri, M. Gumma, O. R. P. Gangalakunta, H. Turrall, X. Cai, J. Vithanage, M. Schull, and R. Dutta. “Global irrigated area map (GIAM), derived from remote sensing, for the end of the last millennium”. *International Journal of Remote Sensing* 30.14 (2009), pp. 3679–3733. DOI: [10.1080/01431160802698919](https://doi.org/10.1080/01431160802698919).
- [9] J. Salmon, M. A. Friedl, S. Frohking, D. Wisser, and E. M. Douglas. “Global rain-fed, irrigated, and paddy croplands: A new high resolution map derived from remote sensing, crop inventories and climate data”. *International Journal of Applied Earth Observation and Geoinformation* 38 (2015), pp. 321–334. DOI: [10.1016/j.jag.2015.01.014](https://doi.org/10.1016/j.jag.2015.01.014).
- [10] A. Puy, S. Lo Piano, and A. Saltelli. “Current models underestimate future irrigated areas”. *Geophysical Research Letters* 47.8 (Apr. 2020). DOI: [10.1029/2020GL087360](https://doi.org/10.1029/2020GL087360).
- [11] S. Siebert, P. Döll, J. Hoogeveen, J.-M. Faures, K. Frenken, and S. Feick. “Development and validation of the global map of irrigation areas”. *Hydrology and Earth System Sciences* 9.5 (Nov. 2005), pp. 535–547. DOI: [10.5194/hess-9-535-2005](https://doi.org/10.5194/hess-9-535-2005).
- [12] J. Bonner. *Why Size Matters*. Princeton and Oxford: Princeton University Press, 2006.
- [13] E. P. White, S. M. Ernest, A. J. Kerkhoff, and B. J. Enquist. “Relationships between body size and abundance in ecology”. *Trends in Ecology & Evolution* 22.6 (June 2007), pp. 323–330. DOI: [10.1016/j.tree.2007.03.007](https://doi.org/10.1016/j.tree.2007.03.007).

- [14] L. M. A. Bettencourt, J. Lobo, D. Helbing, C. Kühnert, and G. West. “Growth, innovation, scaling, and the pace of life in cities”. *Proceedings of the National Academy of Sciences* 104.17 (2007), pp. 7301–7306. DOI: [10.1073/pnas.0610172104](https://doi.org/10.1073/pnas.0610172104).
- [15] G. West. *Scale. The Universal Laws of Growth, Innovation, Sustainability, and the Pace of Life, in Organisms, Cities, Economies and Companies*. New York: Penguin Press, 2017.
- [16] Y. Wada, D. Wisser, and M. F. Bierkens. “Global modeling of withdrawal, allocation and consumptive use of surface water and groundwater resources”. *Earth System Dynamics* 5.1 (2014), pp. 15–40. DOI: [10.5194/esd-5-15-2014](https://doi.org/10.5194/esd-5-15-2014).
- [17] N. Hanasaki, S. Yoshikawa, Y. Pokhrel, and S. Kanae. “A global hydrological simulation to specify the sources of water used by humans”. *Hydrology and Earth System Sciences* 22 (2018), pp. 789–817. DOI: [10.5194/hess-2017-280](https://doi.org/10.5194/hess-2017-280).
- [18] J. Jägermeyr, D. Gerten, J. Heinke, S. Schaphoff, M. Kummu, and W. Lucht. “Water savings potentials of irrigation systems: Global simulation of processes and linkages”. *Hydrology and Earth System Sciences* 19.7 (2015), pp. 3073–3091. DOI: [10.5194/hess-19-3073-2015](https://doi.org/10.5194/hess-19-3073-2015).
- [19] H. Muller Schmied, L. Adam, S. Eisner, G. Fink, M. Florke, H. Kim, T. Oki, F. Theodor Portmann, R. Reinecke, C. Riedel, Q. Song, J. Zhang, and P. Doll. “Variations of global and continental water balance components as impacted by climate forcing uncertainty and human water use”. *Hydrology and Earth System Sciences* 20.7 (2016), pp. 2877–2898. DOI: [10.5194/hess-20-2877-2016](https://doi.org/10.5194/hess-20-2877-2016).
- [20] D. Adam. “Simulating the pandemic: What COVID forecasters can learn from climate models”. *Nature* 587.7835 (Nov. 2020), pp. 533–534. DOI: [10.1038/d41586-020-03208-1](https://doi.org/10.1038/d41586-020-03208-1).
- [21] D. A. Stainforth and R. Calel. “New priorities for climate science and climate economics in the 2020s”. *Nature Communications* 11.1 (2020), pp. 10–12. DOI: [10.1038/s41467-020-16624-8](https://doi.org/10.1038/s41467-020-16624-8).
- [22] D. Wisser, S. Frohking, E. M. Douglas, B. M. Fekete, C. J. Vörösmarty, and A. H. Schumann. “Global irrigation water demand: Variability and uncertainties arising from agricultural and climate data sets”. *Geophysical Research Letters* 35.24 (2008), pp. 1–5. DOI: [10.1029/2008GL035296](https://doi.org/10.1029/2008GL035296).
- [23] S. Siebert, V. Henrich, K. Frenken, and J. Burke. *Update of the digital global map of irrigation areas to version 5*. Rome, 2013.
- [24] Y. Liu, M. Hejazi, P. Kyle, S. H. Kim, E. Davies, D. G. Miralles, A. J. Teuling, Y. He, and D. Niyogi. “Global and regional evaluation of energy for water”. *Environmental Science and Technology* 50.17 (2016), pp. 9736–9745. DOI: [10.1021/acs.est.6b01065](https://doi.org/10.1021/acs.est.6b01065).
- [25] Z. Huang, M. Hejazi, X. Li, Q. Tang, C. Vernon, G. Leng, Y. Liu, P. Döll, S. Eisner, D. Gerten, N. Hanasaki, and Y. Wada. “Reconstruction of global gridded monthly sectoral water withdrawals for 1971–2010 and analysis of their spatiotemporal patterns”. *Hydrology and Earth System Sciences* 22.4 (2018), pp. 2117–2133. DOI: [10.5194/hess-22-2117-2018](https://doi.org/10.5194/hess-22-2117-2018).
- [26] R. V. O’Neill. “Error analysis of ecological models”. *Radionuclides in Ecosystems. Proceedings of the Third National Symposium on Radioecology, May 10-12, 1971, Oak Ridge, Tennessee*. Ed. by D. J. Nelson. Springfield, Vancouver, 1971.

- [27] A. Saltelli. “A short comment on statistical versus mathematical modelling”. *Nature Communications* 10.1 (2019), pp. 8–10. DOI: [10.1038/s41467-019-11865-8](https://doi.org/10.1038/s41467-019-11865-8).
- [28] M. Maslin. “Cascading uncertainty in climate change models and its implications for policy”. *Geographical Journal* 179.3 (2013), pp. 264–271. DOI: [10.1111/j.1475-4959.2012.00494.x](https://doi.org/10.1111/j.1475-4959.2012.00494.x).
- [29] R. S. Lunetta, R. G. Congalton, L. K. Fenstermaker, J. R. Jensen, K. C. McGwire, and L. R. Tinney. “Remote sensing and geographic information system data integration: error sources and research issues”. *Photogrammetric Engineering & Remote Sensing* 57.6 (1991), pp. 677–687.
- [30] E. L. Thompson and L. A. Smith. “Escape from model-land”. *Economics* 13 (2019), pp. 1–15. DOI: [10.5018/economics-ejournal.ja.2019-40](https://doi.org/10.5018/economics-ejournal.ja.2019-40).
- [31] S. Siebert, M. Kummu, M. Porkka, P. Döll, N. Ramankutty, and B. R. Scanlon. “A global data set of the extent of irrigated land from 1900 to 2005”. *Hydrology and Earth System Sciences* 19 (2015), pp. 1521–1545. DOI: [10.13019/M20599](https://doi.org/10.13019/M20599).
- [32] L. Warszawski, K. Frieler, V. Huber, F. Piontek, O. Serdeczny, and J. Schewe. “The Inter-Sectoral Impact Model Intercomparison Project (ISI-MIP): Project framework”. *Proceedings of the National Academy of Sciences* 111.9 (Mar. 2014), pp. 3228–3232. DOI: [10.1073/pnas.1312330110](https://doi.org/10.1073/pnas.1312330110).
- [33] D. P. van Vuuren, J. Edmonds, M. Kainuma, K. Riahi, A. Thomson, K. Hibbard, G. C. Hurtt, T. Kram, V. Krey, J. F. Lamarque, T. Masui, M. Meinshausen, N. Nakicenovic, S. J. Smith, and S. K. Rose. “The representative concentration pathways: An overview”. *Climatic Change* 109.1 (2011), pp. 5–31. DOI: [10.1007/s10584-011-0148-z](https://doi.org/10.1007/s10584-011-0148-z).
- [34] A. Puy, R. Munepeerakul, and A. L. Balbo. “Size and stochasticity in irrigated social-ecological systems”. *Scientific Reports* 7 (Mar. 2017), p. 43943. DOI: [10.1038/srep43943](https://doi.org/10.1038/srep43943).
- [35] A. Puy. “Irrigated areas grow faster than the population”. *Ecological Applications* 28.6 (Sept. 2018), pp. 1413–1419. DOI: [10.1002/eap.1743](https://doi.org/10.1002/eap.1743).
- [36] H. Malano and M. Burton. *Guidelines for benchmarking performance in the irrigation and drainage sector*. Tech. rep. Rome: IPTRID Secretariat, Food and Agriculture Organization of the United Nations, 2001.
- [37] ANCID. *Australian Irrigation Water Provider. Benchmarking Data Report for 2003/2004. Key Irrigation Industry and Performance Indicators*. Tech. rep. April. Australian Government Department of Agriculture, Fisheries and Forestry, 2005.
- [38] W. B. Solley, R. R. Pierce, and H. Perlman. *Estimated Use of Water in the United States in 1995*. Tech. rep. US Geological Survey Circular 1200, 1998, p. 50.
- [39] T. Ivahnenko and J. L. Flynn. *Estimated withdrawals and use of water in Colorado, 2005*. Tech. rep. U.S. Geological Survey Scientific Investigations Report 2010-5002, 2010, p. 61.
- [40] K. Tatsumi and Y. Yamashiki. “Effect of irrigation water withdrawals on water and energy balance in the Mekong River Basin using an improved VIC land surface model with fewer calibration parameters”. *Agricultural Water Management* 159 (2015), pp. 92–106. DOI: [10.1016/j.agwat.2015.05.011](https://doi.org/10.1016/j.agwat.2015.05.011).
- [41] Q. Tang, T. Oki, S. Kanae, and H. Hu. “The influence of precipitation variability and partial irrigation within grid cells on a hydrological simulation”. *Journal of Hydrometeorology* 8.3 (2007), pp. 499–512. DOI: [10.1175/JHM589.1](https://doi.org/10.1175/JHM589.1).

- [42] J. Carpenter and J. Bithell. “Bootstrap confidence intervals: when, which, what? A practical guide for medical statisticians”. *Statistics in Medicine* 19.2 (2000), pp. 1141–1164. DOI: [10.1016/j.trstmh.2007.04.008](https://doi.org/10.1016/j.trstmh.2007.04.008).
- [43] FAO. *Crops and Drops. Making the Best Use of Water for Agriculture*. Tech. rep. Rome: Food and Agriculture Organization of the United Nations, 2002.
- [44] M. S. Kukul and S. Irmak. “Irrigation-limited yield gaps: trends and variability in the United States post-1950”. *Environmental Research Communications* 1.6 (2019), p. 061005. DOI: [10.1088/2515-7620/ab2aee](https://doi.org/10.1088/2515-7620/ab2aee).
- [45] United Nations. *Facts and Figures. Managing Water under Uncertainty and Risk*. Tech. rep. United Nations World Water Assessment Programme, 2012.
- [46] E. Boserup. *The Conditions of Agricultural Growth*. London: George Allen & Unwin Ltd, 1965.
- [47] R. M. Netting. *Smallholders, Householders. Farm Families and the Ecology of Intensive, Sustainable Agriculture*. Stanford: Stanford University Press, 1993.
- [48] Y. Wada, L. P. Van Beek, and M. F. Bierkens. “Modelling global water stress of the recent past: On the relative importance of trends in water demand and climate variability”. *Hydrology and Earth System Sciences* 15.12 (2011), pp. 3785–3808. DOI: [10.5194/hess-15-3785-2011](https://doi.org/10.5194/hess-15-3785-2011).
- [49] M. Hejazi, J. Edmonds, L. Clarke, P. Kyle, E. Davies, V. Chaturvedi, M. Wise, P. Patel, J. Eom, K. Calvin, R. Moss, and S. Kim. “Long-term global water projections using six socioeconomic scenarios in an integrated assessment modeling framework”. *Technological Forecasting and Social Change* 81.1 (2014), pp. 205–226. DOI: [10.1016/j.techfore.2013.05.006](https://doi.org/10.1016/j.techfore.2013.05.006).
- [50] Y. Shen, T. Oki, N. Utsumi, S. Kanae, and N. Hanasaki. “Projection of future world water resources under SRES scenarios: water withdrawal”. *Hydrological Sciences* 53.1 (Feb. 2008), pp. 11–33. DOI: [10.1623/hysj.53.1.11](https://doi.org/10.1623/hysj.53.1.11).
- [51] G. Fischer, F. N. Tubiello, H. van Velthuis, and D. A. Wiberg. “Climate change impacts on irrigation water requirements: Effects of mitigation, 1990-2080”. *Technological Forecasting and Social Change* 74.7 (2007), pp. 1083–1107. DOI: [10.1016/j.techfore.2006.05.021](https://doi.org/10.1016/j.techfore.2006.05.021).
- [52] I. Haddeland, J. Heinke, H. Biemans, S. Eisner, M. Flörke, N. Hanasaki, M. Konzmann, F. Ludwig, Y. Masaki, J. Schewe, T. Stacke, Z. D. Tessler, Y. Wada, and D. Wisser. “Global water resources affected by human interventions and climate change”. *Proceedings of the National Academy of Sciences of the United States of America* 111.9 (2014), pp. 3251–3256. DOI: [10.1073/pnas.1222475110](https://doi.org/10.1073/pnas.1222475110).
- [53] M. I. Hejazi, J. A. Edmonds, and V. Chaturvedi. “Global irrigation demand - a holistic approach”. *Irrigation & Drainage Systems Engineering* 1.3 (2012), pp. 2–5. DOI: [10.4172/2168-9768.1000e106](https://doi.org/10.4172/2168-9768.1000e106).
- [54] P. Döll and S. Siebert. “Global modeling of irrigation water requirements”. *Water Resources Research* 38.4 (2002), pp. 8–1–8–10. DOI: [10.1029/2001WR000355](https://doi.org/10.1029/2001WR000355).
- [55] I. M. Sobol’. “On the distribution of points in a cube and the approximate evaluation of integrals”. *USSR Computational Mathematics and Mathematical Physics* 7.4 (Jan. 1967), pp. 86–112. DOI: [10.1016/0041-5553\(67\)90144-9](https://doi.org/10.1016/0041-5553(67)90144-9).

- [56] I. M. Sobol'. "Uniformly distributed sequences with an additional uniform property". *USSR Computational Mathematics and Mathematical Physics* 16.5 (Jan. 1976), pp. 236–242. DOI: [10.1016/0041-5553\(76\)90154-3](https://doi.org/10.1016/0041-5553(76)90154-3).
- [57] A. Puy, W. Becker, S. L. Piano, and A. Saltelli. "The battle of total-order sensitivity estimators" (Sept. 2020). arXiv: [2009.01147](https://arxiv.org/abs/2009.01147).

Reviewer #3 (Remarks to the Author):

Review of the manuscript "Irrigated areas drive irrigation water withdrawals" for Nature Communications

The authors addressed in a proper way all my comments, and I think most of the other reviewers' comments as well. The paper has improved a lot. The new version is more comprehensive in terms of the short- and long-term implications, on the geographical validation of their model, the clarity of the method section, and the link to food security and climate change.

At this point, I only have some minor comments:

- Do you mean SSP2/rcp26 and SSP2/rcp60?
- What are the 14 combinations referred in Fig. 5? For example the 5 for H08?
- Lines 267-268: The sentence is not clear.
- Line 274: There is no equation number for Jaegermeyr et al.

Reply to the Reviewers of
Irrigated areas drive irrigation water withdrawals
(NCOMMS-20-48716-T)

Arnald Puy, Emanuele Borgonovo, Samuele Lo Piano, Simon A. Levin, Andrea Saltelli

June 13, 2021

Authors' comments are in blue.

Reviewer #1

The response provided to the reviewer's comments is appropriate, and modifications made in the text are up to the mark. I find no significant issues with the revised version of the manuscript. Therefore, I recommended this paper for publication.

Regards,

Ali Ajaz, Ph.D.

Irrigation Specialist

Texas Water Resources Institute

Texas A&M University

Thank you!

Reviewer #3

The authors addressed in a proper way all my comments, and I think most of the other reviewers' comments as well. The paper has improved a lot. The new version is more comprehensive in terms of the short- and long-term implications, on the geographical validation of their model, the clarity of the method section, and the link to food security and climate change.

At this point, I only have some minor comments:

Comment 1. Do you mean SSP2/rcp26 and SSP2/rcp60?

Please kindly see our answer to comment 2 below.

Comment 2. What are the 14 combinations referred in Fig. 5? For example the 5 for H08?

The 14 combinations appear listed in Puy [1, p. 45], which is now cited in the caption of Fig. 5 in the upgraded version of the manuscript. We provide a screenshot and an explanation of these combinations in Fig. R1.

```
# FUTURE IRRIGATION WATER WITHDRAWALS -----  
  
files <- list(  
  "pcr-globwb_miroc5_ewembi_rcp60_2005soc_co2_airrww_global_monthly_2006_2099.nc",  
  "pcr-globwb_miroc5_ewembi_rcp26_2005soc_co2_airrww_global_monthly_2006_2099.nc",  
  "lpjml_miroc5_ewembi_rcp60_2005soc_co2_airrww_global_monthly_2006_2099.nc",  
  "lpjml_miroc5_ewembi_rcp26_2005soc_co2_airrww_global_monthly_2006_2099.nc",  
  "lpjml_miroc5_ewembi_rcp60_rcp60soc_co2_airrww_global_monthly_2006_2099.nc",  
  "lpjml_miroc5_ewembi_rcp26_rcp26soc_co2_airrww_global_monthly_2006_2099.nc",  
  "lpjml_miroc5_ewembi_rcp85_2005soc_co2_airrww_global_monthly_2006_2099.nc",  
  "h08_miroc5_ewembi_rcp60_2005soc_co2_airrww_global_monthly_2006_2099.nc",  
  "h08_miroc5_ewembi_rcp26_2005soc_co2_airrww_global_monthly_2006_2099.nc",  
  "h08_miroc5_ewembi_rcp60_rcp60soc_co2_airrww_global_monthly_2006_2099.nc",  
  "h08_miroc5_ewembi_rcp85_2005soc_co2_airrww_global_monthly_2006_2099.nc",  
  "h08_miroc5_ewembi_rcp26_rcp26soc_co2_airrww_global_monthly_2006_2099.nc",  
  "mpi-hm_miroc5_ewembi_rcp60_2005soc_co2_airrww_global_monthly_2006_2099.nc",  
  "mpi-hm_miroc5_ewembi_rcp26_2005soc_co2_airrww_global_monthly_2006_2099.nc"  
)
```

Figure R1: Combinations of SSP2 and RCPs studied in the paper, retrieved from ISIMIP. Each file includes the name of the Global Model, the Climate Forcing, the projected social and climatic setting, the variable of interest (“airrww”, actual irrigation water use), and the simulation time. Simulations with the “2005” label run on land use patterns (including the extension of irrigation) fixed at their 2005 values. Simulations that do not include the “2005” label assume that irrigated areas change according to the Shared Socioeconomic Pathway 2 (SSP2, “Middle of the Road”). rcp26, rcp60 and rcp85 refer to the Representative Concentration Pathway 2.6, 6 and 8.5 respectively. This is explained also in Fig. S19 of the Supplementary Materials.

Comment 3. Lines 267-268: The sentence is not clear.

We have improved the sentence, which now reads as follows:

That GMs are too complex given the quality of the data available is also suggested by the ambiguity surrounding other aspects of their irrigation module.

Comment 4. Line 274: There is no equation number for Jaegermeyr et al.

Thanks for noticing this. We have added the equation numbers in the new version of the manuscript.